# A Flexible Architecture for Extracting Metro Tunnel Cross Sections from Terrestrial Laser Scanning Point Clouds

Zhen Cao [1], Dong Chen [1,*], Yufeng Shi [1], Zhenxin Zhang [2], Fengxiang Jin [3], Ting Yun [4], Sheng Xu [4], Zhizhong Kang [5] and Liqiang Zhang [6]

1   College of Civil Engineering, Nanjing Forestry University, Nanjing 210037, China; caozhennjsg@gmail.com (Z.C.); yfshi@njfu.edu.cn (Y.S.)
2   Advanced Innovation Center for Imaging Technology, Capital Normal University, Beijing 100048, China; zhangzhx@cnu.edu.cn
3   School of Surveying and Geo-Informatics, Shandong Jianzhu University, Jinan 250101, China; fxjin@sdjzu.edu.cn
4   College of Information Science and Technology, Nanjing Forestry University, Nanjing 210037, China; njyunting@gmail.com (T.Y.); xusheng404@gmail.com (S.X.)
5   School of Land Science and Technology, China University of Geosciences, Beijing 100083, China; zzkang@cugb.edu.cn
6   The State Key Laboratory of Remote Sensing Science, Faculty of Geographical Science, Beijing Normal University, Beijing 100875, China; zhanglq@bnu.edu.cn
*   Correspondence: chendong@njfu.edu.cn

**Abstract:** This paper presents a novel framework to extract metro tunnel cross sections (profiles) from Terrestrial Laser Scanning point clouds. The entire framework consists of two steps: tunnel central axis extraction and cross section determination. In tunnel central extraction, we propose a slice-based method to obtain an initial central axis, which is further divided into linear and nonlinear circular segments by an enhanced Random Sample Consensus (RANSAC) tunnel axis segmentation algorithm. This algorithm transforms the problem of hybrid linear and nonlinear segment extraction into a sole segmentation of linear elements defined at the tangent space rather than raw data space, significantly simplifying the tunnel axis segmentation. The extracted axis segments are then provided as input to the step of the cross section determination which generates the coarse cross-sectional points by intersecting a series of straight lines that rotate orthogonally around the tunnel axis with their local fitted quadric surface, i.e., cylindrical surface. These generated profile points are further refined and densified via solving a constrained nonlinear least squares problem. Our experiments on Nanjing metro tunnel show that the cross sectional fitting error is only 1.69 mm. Compared with the designed radius of the metro tunnel, the RMSE (Root Mean Square Error) of extracted cross sections' radii only keeps 1.60 mm. We also test our algorithm on another metro tunnel in Shanghai, and the results show that the RMSE of radii only keeps 4.60 mm which is superior to a state-of-the-art method of 6.00 mm. Apart from the accurate geometry, our approach can maintain the correct topology among cross sections, thereby guaranteeing the production of geometric tunnel model without crack defects. Moreover, we prove that our algorithm is insensitive to the missing data and point density.

**Keywords:** terrestrial laser scanning; tunnel central axis; tunnel cross section; enhanced RANSAC; quadric fitting; constrained nonlinear least-squares problem

## 1. Introduction

　　Tunnel displacement and deformation monitoring are critical factors for engineers to evaluate the tunnel health and safety. Some indices such as the cross section's chord [1], convergence and dislocations [2], tunnel axis's settlement [2], and tunnel model's clearance [3] are important measures to evaluate the stability of tunnels. At the initial stage, some traditional methods are used to monitor the tunnel's health. The instruments such as tape extensometer [4] and total station [5] are extensively used for high precision deformation monitoring. Although they have a very high precision, these traditional ground surveying measurements are time-consuming, labor-intensive and costly. In some cases, they fail to get the deformation because sparsely measured points are not dense enough to evaluate the overall tunnel. Apart from the above geodetic surveys, an alternative method based on digital photogrammetry has been proposed because more measured points along a profile can be extracted efficiently. However, many of the acquired images easily suffer from occlusions, low contrast and unfavorable perspectives, thereby making automatic image matching and understanding harder. For example, by combining stereo images and laser-lit spots, Wang et al. [5,6] propose a profile-image method to measure cross sections. Although this method captures the densified points along a profile more than traditional geodetic surveys, it is restricted by insufficient lighting conditions in an actual tunnel. We recommend the readers to the recent survey [7] to get the review of the state-of-the-art methods and applications in tunnel inspections based on photogrammetric and computer vision techniques. Recently, the Terrestrial Laser Scanning (TLS) technique has been developed to efficiently and accurately collect three-dimensional point clouds from the reflected objects. Since the advantages of TLS technique include being highly accurate, efficient, automatic and has a lack of contact, the TLS technique has been proved to be an effective way to monitor the deformation of engineering structures, particularly in the fields of tunnel management [8,9] and deformation analysis [10–12].

　　Based on TLS point clouds, tunnel deformation monitoring techniques can be generally categorized into profile-based methods and model-based methods. The profile-based method tries to analyze the tunnel's deformation by using a series of thin profiles directly derived from TLS tunnel point clouds. Numerous published papers along this line [1,13–15] demonstrate the accuracy and efficiency of this type of method. For example, Han et al. [13] propose an automated and efficient method to estimate the tunnel centerline and cross-sectional plane. In their method, the centerline is first estimated by skeletonizing a binary image produced by projecting three-dimensional point clouds onto a horizontal plane. Based on the tunnel boundary points and the centerline, the cross-sectional planes are estimated and further adjusted, and the final cross sections are generated by projecting the nearby points to the adjusted planes. Although the authors improve the efficiency of surveying and data processing, they cannot extract the continuous profiles because the parametric equation of tunnel centerline is not provided. In addition, this method is sensitive to the non-lining points, i.e., pipes and equipment attached to the lining. To overcome the drawback regarding the above centerline, the methods in [1,3] use multiple models, i.e., straight line, transition curve and curve to commonly fit the different segments of tunnel centerline, and then adopt the global least-squares adjustment to maintain the consistency between adjacent fitting models. To cope with missing data, Kang et al. [1] recover the blank areas of tunnel profiles with a surficial interpolation algorithm. To obtain the accurate direction of the cross-sectional planes, the authors in [14] optimize the initial cross sections by means of twice adjustments using the total least squares method and Rodrigues' rotation formula, respectively.

　　The critical step for profile-based methods is to obtain the real cross-sectional points. To this end, some methods focus on how to eliminate non-lining points (accessories) from the raw point clouds. Generally, two types of methods are used to remove non-lining points. The first method is based on the geometric characteristics of the tunnel. More specifically, it is assumed that the tunnel can be fitted by the circular model [1,11,12,14,15] and then uses the circular filtering to filter cables and other equipment in the tunnel. The second method separates liner and accessories by combining TLS geometric and radiometric information. For example, the results in [16] demonstrate that the corrected intensity information is an effective physical criterion and a complementary data source to remove

accessories that cannot be eliminated by sole geometric information. In another method, Yoon et al. [8] use the radiometric characteristics for Mobile Laser Scanning (MLS) data to filter the accessories from the tunnel raw data, which cannot be directly used for processing the TLS point clouds because not only the scattering property of an object but also other variables, especially distance and incidence angle effects will influence backscattered energy in TLS data [3]. To address this deficiency, Tan et al. [17] further investigate the effect of distance on the intensity of TLS data and propose a new correction method for different TLS scanners to overcome the deficiency of directly using the original intensity.

Although the acquisition of actual geometric parameters for inspection of tunnel is efficient, accurate and has flexible implementation from TLS point clouds, the profile-based works are more restricted regarding pointwise and piecewise description of the profiles. That is, the profile-based method has some potential risks between the areas of adjacent cross sections, where the slight deformations might not be tracked. To address this issue, the profile-based method has been expanded to an overall evaluation of tunnel models. For example, Fekete et al. [9] reconstruct the models by means of triangulating tunnel point clouds to detect the underbreak and overbreak areas. Dimitrov and Golparvar-Fard [18] use the non-uniform rational B-spine (NURBS) fitting technique to achieve parametric representation of arbitrary 3D geometries from point clouds. Nahangi and Haas [19] present an algorithm for automated quantification of discrepancies for components of assemblies by using skeletonized 3D models and TLS laser scans. Qiu and Cheng [3] propose a novel clearance inspection technique to generate the high-resolution digital surface model of a railway tunnel surface (bare lining) from TLS point clouds. By introducing a high-accuracy interpolation and filtering algorithm, this method can generate bare-lining models without limitation of tunnel cross section shapes. In Ref. [20], the scanned surface of a bored tunnel is approximated with a cylindrical model, and the point-wise deformation analysis is performed by comparing surface patches. Although the tunnel model-based method can detect the minor differences by comparison with the tunnel models generated at different times, this type of method is sensitive to non-lining point clouds, which means the real deformation and displacement can be hidden behind the artifacts of the created tunnel models. In addition, the generation of tunnel models ranging from geometric primitives, e.g., planes, cylinder or spheres, to more complex ones, such as parametric patches and NURBS is time-consuming, which does not meet the requirement of real-time interaction for processing large-scale TLS point clouds.

In this paper, we employ the hybrid concept by combining the profile-based and model-based methods for tunnel deformation monitoring. More specifically, based on the concept of the profile-based method, we propose a novel cross-sectional extraction method, which not only gets the continuous points within each cross section, but also more importantly has the capability to generate continuous profiles over the entire metro tunnel. Utilizing these continuous points and profiles, the entire tunnel model can be easily generated by triangulation. Thus, if we have two-phase TLS tunnel data acquired at different times, we can evaluate the whole differences by the superposition of two phase tunnel models. Given that our work is built on profile-based framework, we explicitly state our original contributions as follows:

- **Tunnel Central Axis Extraction:** we present an algorithm for tracking the tunnel central axis in topologically-ware and geometrically-aware manners. That is, we first propose a slice-based algorithm for tracking the initial central axis, which is further refined by integrating a Minimum Spanning Tree (MST) and Savitzky–Golay smoothing.
- **Tunnel Central Axis Segmentation:** we propose an enhancement to the traditional Random Sample Consensus (RANSAC) algorithm. The enhanced algorithm translates the problem of hybrid linear and nonlinear segment extraction into a sole segmentation of linear elements defined at the tangent space rather than raw data space, significantly simplifying segmentation problem.
- **Tunnel Profile Determination:** we introduce a high-accuracy interpolation and filtering algorithm for extracting continuous tunnel profiles over the entire tunnel. The generated profile points are determined by using interpolation, and further densified via solving a constrained nonlinear least squares formulation.

This paper is organized as follows: Section 1 reviews profile-based and model-based methods in tunnel deformation. Section 2 describes the detailed methodology including tunnel centerline extraction and cross section determination. In Section 3, the experimental dataset, the performance evaluation results of geometric and topological accuracy based on actual TLS tunnel data are presented, analyzed, and discussed. Finally, Section 4 concludes the paper along with a few suggestions for future research topics.

## 2. Methodology

### 2.1. Tunnel Axis Extraction via the Slice-based Method

The tunnel axis determines the tunnel's position and orientation, and meanwhile reflects the distribution of the tunnel points in three-dimensional space. The tunnel axis is also a basis for extraction of a cross-sectional plane at a specific location because the tunnel's central axis needs to be orthogonal to all of the cross sections. To accurately extract the tunnel axis, in this paper, we introduce a slice-based method, which transforms the problem of extracting a three-dimensional tunnel axis into extracting a two-dimensional axis problem at two projected spaces, i.e., $xy$- and $xz$-planes. That is, we use two-axis equations in two-dimensional space to commonly represent the tunnel axis equation in three-dimensional space.

To get the tunnel axis in the projected $xy$-plane, the projected point clouds are first divided into a series of slices at equal intervals along the direction of $x$-axis. In each slice, the minimum and maximum values on $y$-axis are determined. The mean point calculated by averaging the minimum and maximum points is regarded as an axis point within the corresponding slice. All of the mean points constitute the complete tunnels' axis points in the projected $xy$-plane. As shown in Figure 1a, the point clouds are partitioned into a series of slices by the dash lines at a given interval $\delta$. The point $\bar{p}_i$ is an estimated tunnel axis point calculated by averaging the points $p_i^{max}$ and $p_i^{min}$ within $i$-th slice. Using the above method, all of the points denoted in red are extracted, representing the final tunnel axis points in the projected $xy$-plane. Similarly, the tunnel axis' points are extracted from the projected $xz$-plane as evident in Figure 1b.

It should be noted that the calculated axis points in the beginning and ending areas might deviate from the real position of a tunnel axis due to irregular shapes of the raw data. As these pseudo axis points indicated by red ovals in Figure 1 undoubtedly weaken the accuracy of the extracted tunnel axis, they are eliminated using a local curvature measure. After that, the axis points in the $xy$-plane are fitted by a fifth degree polynomial regression model, i.e., $f_{xy}(x) = \sum_{i=0}^{n} a_i x^i$. Similarly, axis points in the $xz$-plane are fitted by a linear regression model, i.e., $f_{xz}(x) = ax + b$ (see Section 2.2.3). Therefore, the tunnel axis in three-dimensional data space can be represented through joining these two models together:

$$\mathcal{L} = \begin{cases} y = f_{xy}(x), \\ z = f_{xz}(x). \end{cases} \tag{1}$$

Compared with the designed location of the tunnel axis, the height of the tunnel axis in the $xz$-plane has been raised and is usually higher than the theoretical (designed) value. This discrepancy is that, when we scan tunnel point clouds, the track laying has been finished, thereby raising the bottom of a tunnel. Their geometric relationships can be vividly shown in Figure 2, from which the extracted axis point $o$ in red is higher than a center of real tunnel, denoted by a black point $o'$.

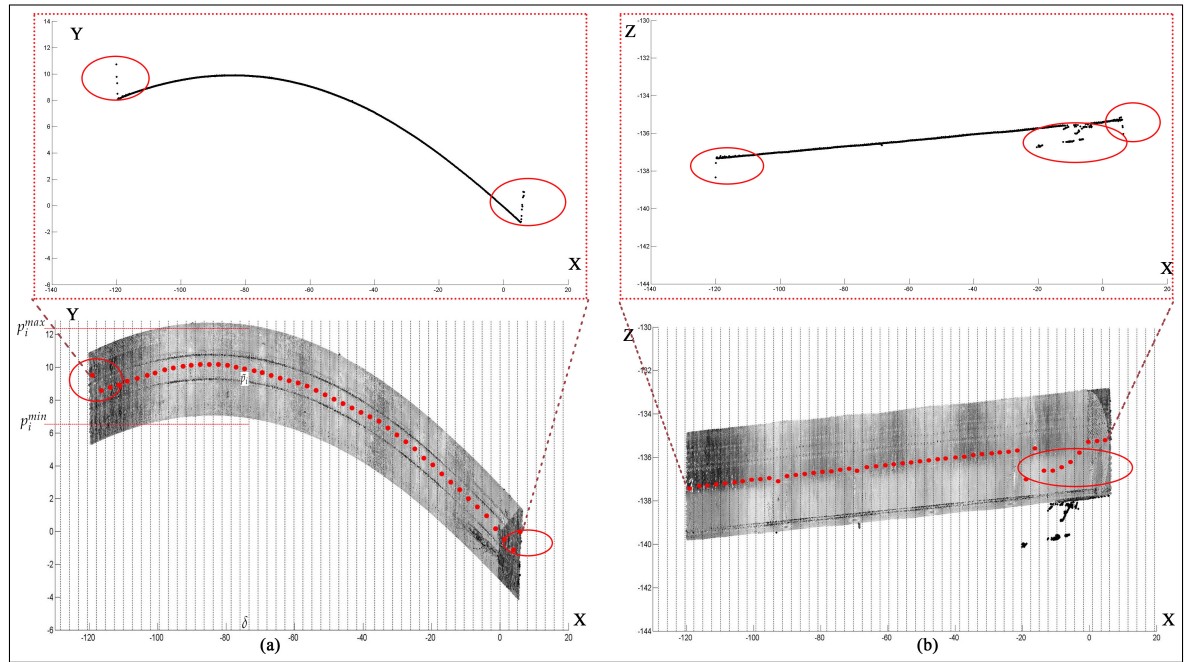

**Figure 1.** Extraction of tunnel axis in the two projected spaces. (**a**) tunnel axis points in $xy$-plane; (**b**) tunnel axis points in the $xz$-plane. The gray points represent the raw tunnel point clouds and the red points represent the calculated tunnel axis points.

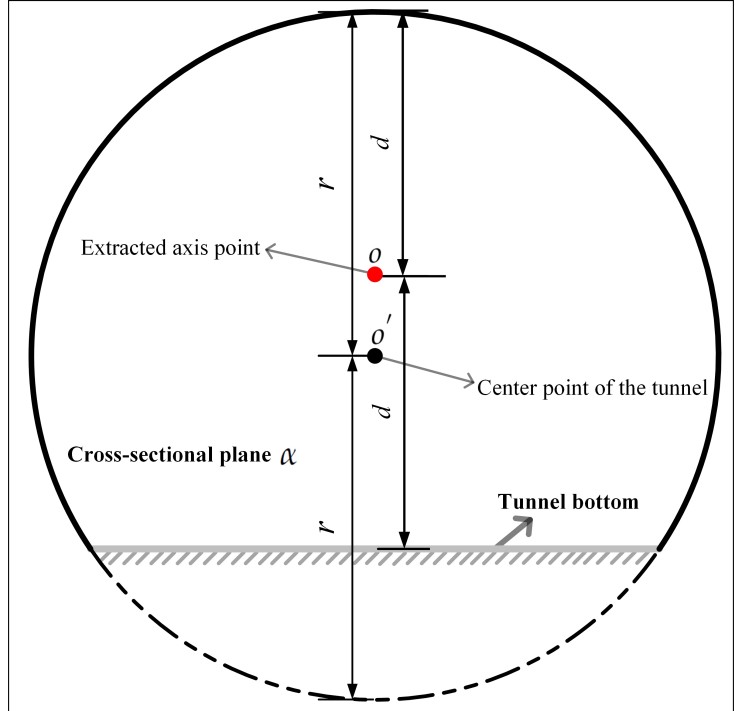

**Figure 2.** Geometric relationships between the designed and the extracted axes.

### 2.2. Tunnel Axis Segmentation

According to the planning requirements regarding tunnel safety, the tunnel generally includes linear parts and nonlinear/circular segments with fixed curvature [3]. Once the tunnel axis has been extracted, it should be further segmented into linear and nonlinear segments. For the linear segments, we refit them by the linear least squares regression. However, we use a high order polynomial model to fit the nonlinear segments. The advantage of axis segmentation is twofold: (1) The tangent and

directional vectors corresponding to circular and linear segments can be easily determined. This means the axis direction for any point along the axis can be estimated, which prepares the conditions for the subsequent extraction of cross sections as the tunnel axis should be orthogonal to the cross sections. (2) The axis segmentation can help us to achieve the tunnel's geometric representation at multiple levels of details (LoDs). More specifically, for linear segments, a few tunnel cross sections are enough to fit/represent the tunnel geometry. In contrast, for circular segments or the transition zones between linear and circular segments, the high density of cross sections are needed to achieve an accurate geometric representation. Through tunnel axis segmentation, we can enhance the flexibility of geometric representation via using the concept of LoD representation. Meanwhile, we can also make a good balance between the tunnel's geometric accuracy and compactness via using a few tunnel cross sections.

The proposed axis segmentation algorithm consists of three stages: (1) We first restore the topological relationships of tunnel axis points using MST. (2) The sequence axis points are then provided as input to the smoothing step which uses the Savitzky–Golay (SG) [21] filtering algorithm to refine the axis. (3) In the final step, the axis has been divided into linear and nonlinear segments using an enhanced RANSAC [22] algorithm, which works at the tangent space rather than the data space defined in the traditional sense.

### 2.2.1. Constructing MST of Point Clouds along the Tunnel Axis

The initial extracted axis points in Section 2.1 are unorganized and without topology. We need to restore the topological relationships, i.e., clock-wise or counterclockwise order of these points. To this end, we first construct the MST, from which the weight on any edge is the Euclidean distance between two points. After the completion of the MST, the maximum depth of a subtree is traversed via the depth-first searching (DFS) algorithm. This subtree implicitly includes the topological relationship among axis points, i.e., clockwise or counterclockwise sequence of axis points. As shown in Figure 3a, the central axis of a tunnel can be well recognized via undirected graph MST although it has a certain degree of noises along the central axis. After DFS traversing, in Figure 3b, the sequence of the central axis points connected by the green lines is well maintained. Apart from restoring topology of axis points, it can be observed in Figure 3b, our MST-based traversing strategy has the capability to remove a certain degree of noises which deviates far from the tunnel axis to some extent, thereby refining the axis points.

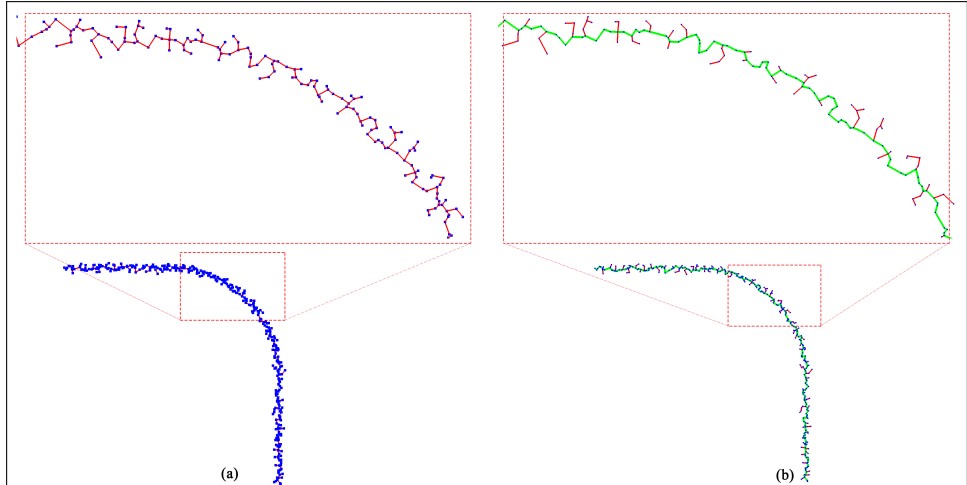

**Figure 3.** Restoring the topological relationship among axis points via MST (Minimum Spanning Tree). (**a**) the axis point clouds are organized by an undirected graph MST; (**b**) the maximum depth of a subtree is recognized and overlaid on the MST. The blue points present the initial axis points. The red and green lines represent the established MST and the extracted axis, respectively.

### 2.2.2. Smoothing Central Axis of a Tunnel

To obtain more accurate axis segments, the central axis acquired in Section 2.2.1 needs to be further smoothed to depress the noises. Fortunately, we find that the SG smoothing filtering is an ideal tool to strike a balance between maintenance of axis details and depression of noise. Essentially, this algorithm performs a local least squares polynomial regression of degree $k$ over at least $k + 1$ points.

Through SG smoothing, the linear and nonlinear patterns/characteristics of the central axis become more obvious and it can be easily recognized in the subsequent segmentation algorithm. As shown in Figure 4, we compare the distribution of points along the central axis in the tangent space, i.e., $(d, \theta)$ plane. Parameter $d$ represents the sum of the lengths of any two connected points along a central axis limited from the beginning point to the current processing point, and $\theta$ represents the angle between the tangent to the current points and the horizontal axis defined in the tangent plane. Obviously, a linear segment of a tunnel is expected to be transformed into a horizontal linear segments in the tangent space. However, a circular segment with a fixed curvature is anticipated to be transformed into a sloping linear segment because the angle between tangents at endpoints of the circular arc is proportional to the length of the arc. In practice, even though we perform the axis segmentation in tangent plane, the axis points contain too many details and "zigzag" noises because irregularly spaced axis point measurements can blur the horizontal and the sloping linear characteristics in tangent space.

We use two types of the geometric tunnel axis based on synthetic points to prove the applicability of the SG algorithm. We adopt the synthetic data rather than the real tunnel point clouds because the Nanjing tunnel dataset used in our paper has a small curvature, which directly results in the curvature of the extracted tunnel axis is not prominent in the projected $xy$-plane. Because of this, the real tunnel point clouds cannot fully test the applicability of the SG filtering algorithm and its influences for the follow-up tunnel axis segmentation algorithm. As shown in Figure 4a, two kinds of geometric tunnels axis points without noise are transformed into tangent plane, where the linear features are prominent. However, after adding a certain degree of noise, the linear distribution patterns in $(d, \theta)$ space are unrecognizable as proved in Figure 4b. After further performing the SG smoothing, the linear distribution of axis points in Figure 4c becomes notable again although it is not as prominent as the situation in Figure 4a.

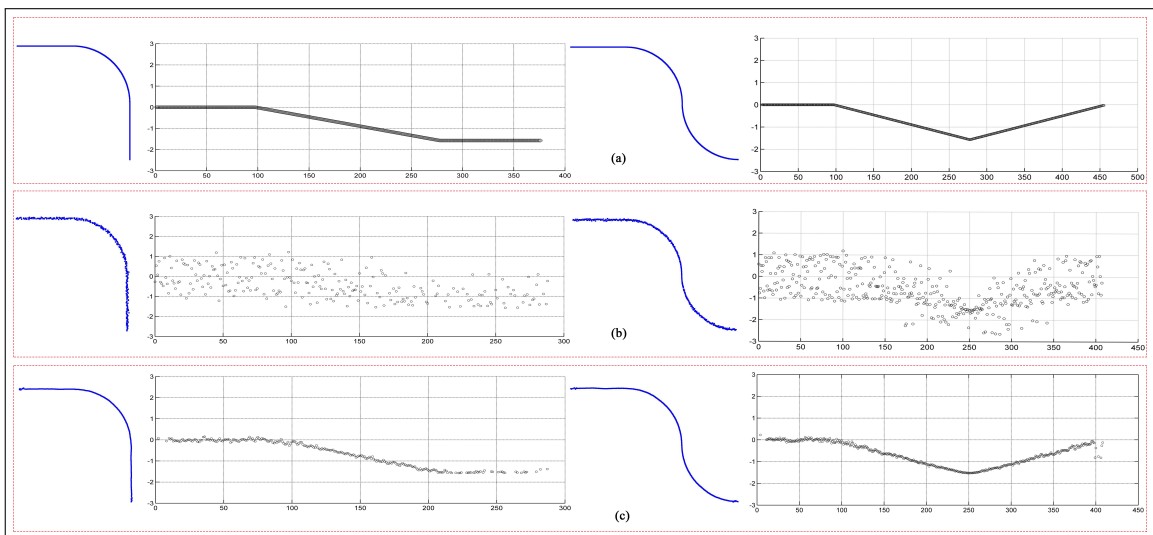

**Figure 4.** Comparisons of linear distributions of axis points in tangent space before and after SG (Savitzky-Golay) smoothing. (**a**) the ideal tunnel axis points without noise in their data and tangent spaces; (**b**) some degree of Gaussian noises have been added into the axis point clouds; (**c**) the distribution of axis points in data and tangent spaces after SG smoothing.

### 2.2.3. Segmentation of the Tunnel Axis

We employ an enhanced RANSAC algorithm to partition the central tunnel axis into linear and circular segments. If the conventional RANSAC algorithm is simply used to simultaneously recognize linear and circular segments in data space, over-segmentation often occurs. For example, a circular segment with low curvature might be mistakenly segmented as multiple linear segments. To solve this problem, we transform the question of detecting the linear and circular segments from the original data space into the question of only detecting linear segments in the transformed tangent space, significantly reducing the complexity of segmentation of the tunnel axis. An enhanced RANSAC algorithm is described as follows:

(a)　The central axis point clouds are transformed into tangent space $(d, \theta)$.

(b)　An optimal linear segment is first detected from $(d, \theta)$ space, and then the axis point clouds are divided into inliers and outliers.

(c)　We determine the extent of inliers in tangent space. If any point from outliers is inside the extent of the inliers, the corresponding point is removed from the outliers.

(d)　The point clouds from the remaining outliers are provided as input. The steps from (*b*) to (*c*) are executed repeatedly until all of the remaining outliers have been processed.

(e)　The sequence of linear elements extracted in tangent space is restored according to the sequence of their middle points. Hence, we obtain anchor points by circulating around pairwise neighboring linear segments. More precisely, if a linear segment $\mathcal{S}_i$ is a neighbor with a linear primitive $\mathcal{S}_j$, there is an anchor point whose position is exactly at the intersection of $\mathcal{S}_i$ and $\mathcal{S}_j$.

(f)　Using these anchor points, the linear and circular segments in data space are simultaneously determined, since the axis points have a one-to-one mapping relationship between data and tangent spaces.

It should be aware that we only need two parameters to execute the enhanced RANSAC algorithm: (1) the probability $P$ that at least one of the selected axis points does not contain an outlier, and (2) the Euclidean distance $\epsilon$ from a point to the hypothetical linear model that determines the number of points consistent with the linear model in the tangent space. In our case, we set parameter $P$ to 0.99, which ensures the optimal linear model within a specific number of iterations controlled by $P$ can be obtained. The other parameter $\epsilon$ is determined by the definition of the tangent space. We set $\epsilon$ to 0.01. It is to be noted that the parameter of $\epsilon$ in the tangent space is dimensionless although its unit is consistent with the unit of raw point clouds in original data space. Figures 5 and 6 show the tunnel axis segmentation results with regard to synthetic and real data. It can be seen that the proposed method can effectively segment the linear and nonlinear axis geometry by introducing the tangent space. Meanwhile, thanks to the SG filter, it makes our axis segmentation insensitive to the noises.

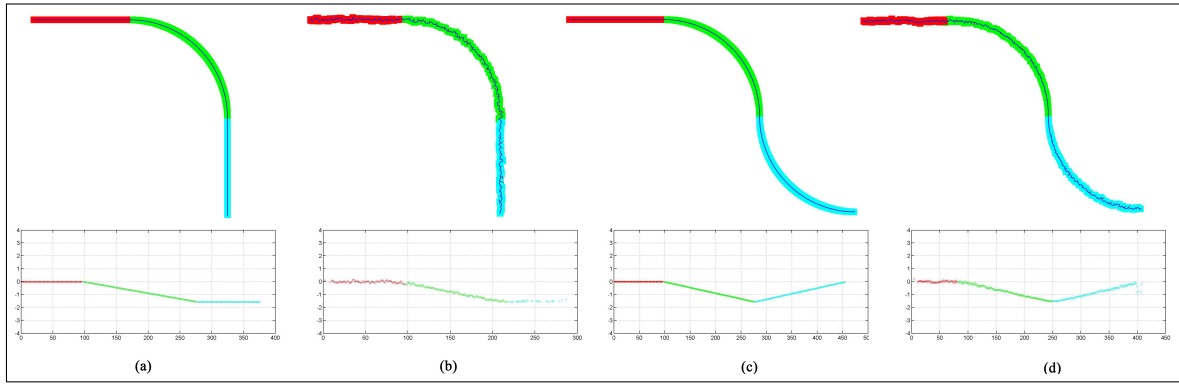

**Figure 5.** Comparisons of tunnel axis segmentation with and without noise. (**a**,**c**) are results of tunnel axis point clouds without noise; (**b**,**d**) are results of the tunnel axis after adding a certain degree of Gaussian noise.

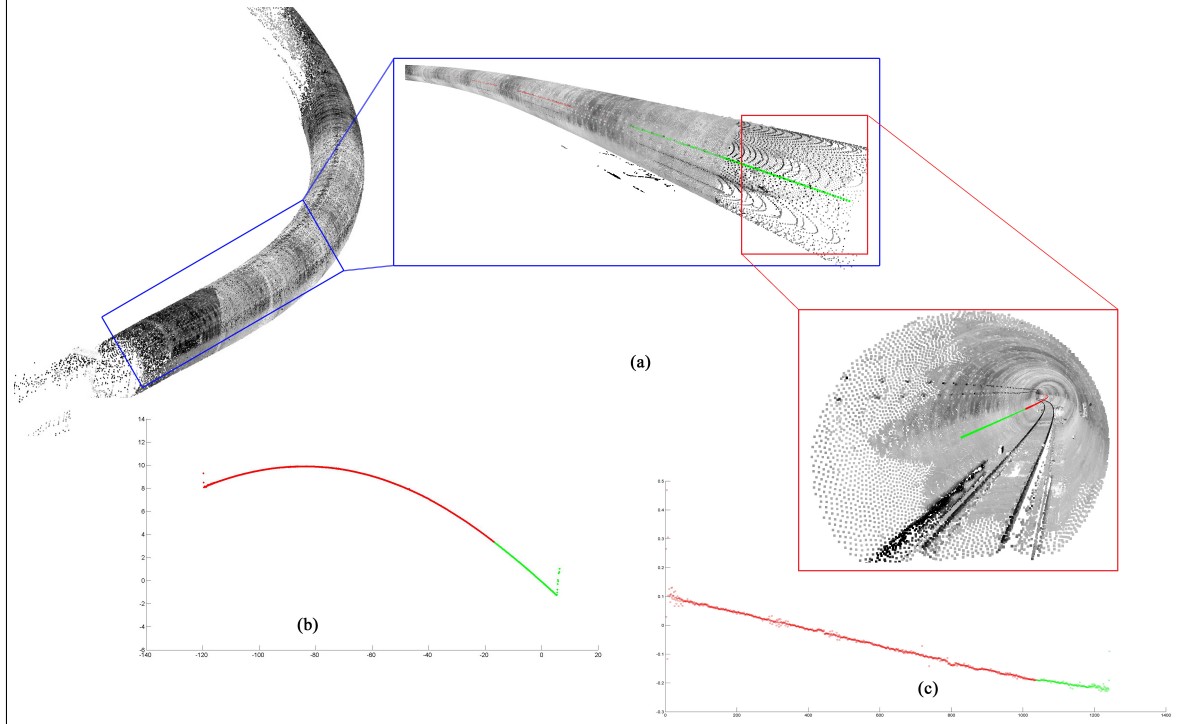

**Figure 6.** The central tunnel axis segmentation of real data. (**a**) the segmented three-dimensional axis is superimposed onto their associated tunnel point clouds; (**b**,**c**) represent the two-dimensional segmented tunnel axis in data space (*xy*-plane) and tangent space, respectively.

After segmentation, we use the linear regression model to fit the linear axis model. The polynomial regression model is employed to fit the axis points for nonlinear tunnel segments. The details are described below:

Generally, the polynomial of degree $n$ can be written into:

$$y = a_0 + a_1 x + a_2 x^2 + \cdots + a_n x^n. \tag{2}$$

As the variable $x$ is the observation value, we let $X_1 = x, X_2 = x^2, \ldots, X_n = x^n$. The nonlinear Equation (2) can be transformed into the linear equation below:

$$y = a_0 + a_1 X_1 + a_2 X_2 + \cdots + a_n X_n. \tag{3}$$

We assume that there are *m* points on the nonlinear axis. Based on Equation (3), these points constitute a set of equations:

$$
\begin{cases}
y_1 = a_0 + a_1 X_{11} + a_2 X_{12} + \ldots a_n X_{1n}, \\
y_2 = a_0 + a_1 X_{21} + a_2 X_{22} + \ldots a_n X_{2n}, \\
\qquad\qquad \ldots \\
y_m = a_0 + a_1 X_{m1} + a_2 X_{m2} + \ldots a_n X_{mn}.
\end{cases}
\tag{4}
$$

Equation (4) can be further transformed into a matrix equation:

$$
Y = BA,
\tag{5}
$$

where $Y = [y_1, y_2, \ldots, y_n]^T$, $B = \begin{bmatrix} 1 & X_{11} & X_{12} & \ldots & X_{1n} \\ 1 & X_{21} & X_{22} & \ldots & X_{2n} \\ \vdots & \vdots & \vdots & \vdots & \vdots \\ 1 & X_{m1} & X_{m2} & \ldots & X_{mn} \end{bmatrix}$, and $A = [a_0, a_1, \ldots, a_n]^T$. $Y$ and $B$ are estimated from the observed data. Equation (5) can be solved using the linear least squares techniques:

$$
Y + e = (B + E_B)A,
\tag{6}
$$

where $e$ and $E_B$ are the error matrix of $Y$ and $B$, respectively. In linear algebra, the coefficients of matrix $A$ can be solved through the singular value decomposition (SVD) [23,24] of an augmented matrix $[BY]$.

### 2.3. Tunnel Cross Section Extraction

It consists of four steps to extract cross-sectional points at an arbitrary point *o* along the tunnel axis: (1) We first calculate the cross-sectional plane $\alpha$ at an arbitrary point *o* on the tunnel axis. (2) We generate a series of straight lines that rotate orthogonally around the tunnel axis within $\alpha$ at a specific rotation interval $\Delta\varphi$. (3) According to each straight line $l_i$, we determine its associated local tunnel point set, which is further fitted by a cylinder model. That is, $l_i$ can pass through the region determined by the local tunnel point clouds. (4) The cross-sectional points are then produced by intersecting a series of straight lines with their associated cylinder surfaces.

### 2.3.1. Determination of the Cross-Sectional Plane

Based on Equation (1), we can get a directional vector of an arbitrary point $o = \{o_x, o_y, o_z\}$ on the tunnel axis $\mathcal{L}$, i.e., $\vec{n}_o = \{1, f'_{xy}|_{x=o_x}, f'_{xz}|_{x=o_x}\}$. Obviously, the cross section that passes through *o* can be determined by $\alpha = \vec{op} \cdot \vec{n}_o$, from which the symbol *p* is an arbitrary point that is different from a point *o* on plane $\alpha$. Symbol "·" is a dot product of two vectors. It should be noted that, for a linear segment, $\vec{n}_o$ is a directional vector of the fitted tunnel's axis segment. For a nonlinear segment, $\vec{n}_o$ is a tangent vector of the fitted nonlinear axis segment on point *o* (see Section 2.2.3).

### 2.3.2. Determination of a Straight Line Equation

This step aims to partition cross-sectional plane $\alpha$ according to a given rotation interval angle $\Delta\varphi$. After being partitioned, a series of straight lines can be obtained. Actually, the number of straight lines equals the number of cross-sectional points. As shown in Figure 7a, to get an arbitrary straight line equation from $\alpha$, we select an arbitrary point *p* from $\alpha$, and a new vector $\vec{l}$ from a plane $\alpha$ can be determined via $\vec{l} = \vec{op}$. Another different vector $\vec{m}$ from a plane $\alpha$ can also be simultaneously determined via $\vec{m} = \vec{n_o} \times \vec{l}$, where the symbol "×" represents a cross product of two vectors. Based on the two vectors mentioned above $\vec{l}$ and $\vec{m}$, an arbitrary vector $\vec{k}$ on a plane $\alpha$ can be represented by a linear combination of the normalized vectors $\vec{l}$ and $\vec{m}$. That is,

$\vec{k} = (\vec{l}/|\vec{l}|)\cos\varphi + (\vec{m}/|\vec{m}|)\sin\varphi$. A straight line equation, i.e., $\mathcal{L}_k = \frac{x-o_x}{\vec{k}_x} = \frac{y-o_y}{\vec{k}_y} = \frac{z-o_z}{\vec{k}_z}$ is solely

determined through vector $\vec{k} = \{\vec{k}_x, \vec{k}_y, \vec{k}_z\}$ and $\vec{o}$.

### 2.3.3. Determination of Cross-Sectional Points

This step aims to determine all of the cross section points on the plane $\alpha$. To this end, we first determine the corresponding local tunnel point clouds regarding each straight line $\mathcal{L}_k$. Then, the local tunnel point clouds are fitted by the nonlinear cylinder surface model. The intersection points between $\mathcal{L}_k$ and the cylinder surface are obtained and the potential pseudo intersection point is further removed under the constraint of local tunnel point clouds' data extent.

More precisely, we first get a point $q_k$ from a straight line $\mathcal{L}_k$ and $q_k$ is subject to the constraint $q_k = o + r\vec{k}$. As shown in Figure 7b, it is obvious that all of the $q_k$ from the straight lines constitute the circle with center $o$ and radius $r$. As we previously mentioned in Section 2.1, the centers $o$ and $o'$ do not overlap, which means that the path of $q_k$ is not consistent with the real tunnel surface. Once we determine each $q_k$, the neighborhood searching centered at $q_k$ is performed. The local point set $\mathcal{Q}_k$ indicated by green ovals are expected to be obtained. It is noted that the number of points within $\mathcal{Q}_k$ is determined by $\rho \times \mathcal{A}$, from which $\mathcal{A}$ is the area of searching extent and $\rho$ is the density of tunnel point clouds. To improve the computational efficiency of the neighborhood searching, the K-D is established using the tunnel points in advance. For each local point set $\mathcal{Q}_k$, the conventional RANSAC algorithm is used to fit a cylinder surface whose parameters include an axis of a cylinder, an arbitrary point from the axis, and a radius of a cylinder. The intersection points between $\mathcal{L}_k$ and its corresponding cylinder surface can be calculated on the plane $\alpha$. It should be aware that, for each straight line, two intersection points can be obtained, but one of them is a pseudo point. In this case, the pseudo point should be eliminated under the constraint of local tunnel points' extent. As a result, the real intersection point remains and can be regarded as one of the tunnel's cross-sectional point corresponding to $\mathcal{L}_k$. The above process is executed repeatedly until all of the cross-sectional points on the plane $\alpha$ are calculated.

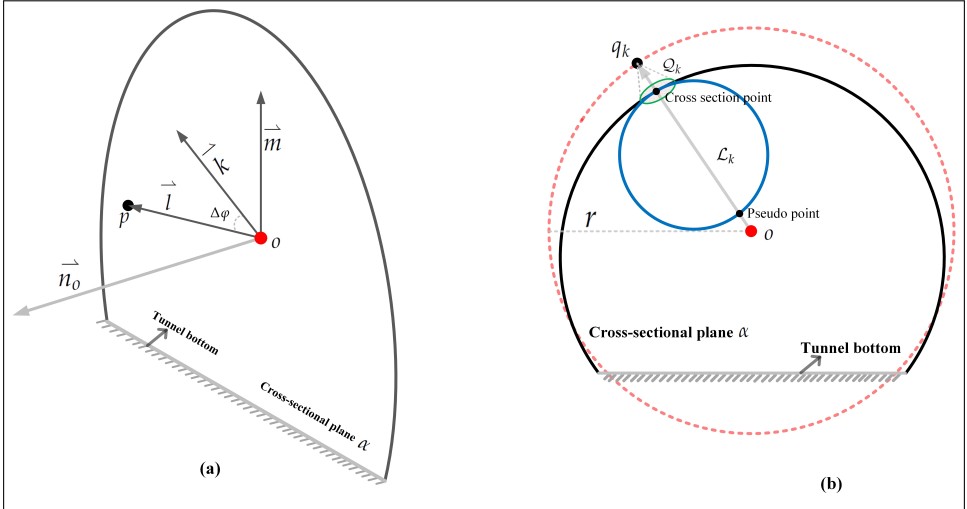

**Figure 7.** The diagram of determination of cross-sectional planes and points. (**a**) cross-sectional plane determination; (**b**) the geometric relationship for determination of cross-sectional points. Note that the dashed circle in red represents the auxiliary circle centered at axis point $o$, while the black circle denotes the real tunnel profile shape centered at point $o'$. The blue circle represents the fitted cylinder using the local point set in $\mathcal{Q}_K$.

In practice, it can be observed that some extracted cross sections have a degree of missing data as it is evident in Figure 8b. That is, some points on the specific cross sections are missing, causing the

cross section's points to be incomplete. This deficiency is mainly because the point set $\mathcal{Q}_k$ includes outliers and noises, or probably because the ratio of the number of liner points in $\mathcal{Q}_k$ is not sufficient to fit a cylinder surface. To further refine the extracted cross sections of a tunnel, we adopt a strategy of "fitting-and-resampling". More specifically, we refit the points of each cross section via the circular model and then get the complete cross-sectional points at a specific rotation interval $\Delta\varphi$ through resampling from the fitted tunnel circles.

To get the parameters of the fitted circle, the least squares regression method under the constraint that the new generated center of the circle must lie on the cross-sectional plane $\alpha$. We assume the circle center $\bar{o} = (\bar{o}_x, \bar{o}_y, \bar{o}_z)$ with radius $\bar{r}$ on the plane $\alpha$. The equation of the circle can be expressed below:

$$x^2 + y^2 + z^2 = 2x\bar{o}_x + 2y\bar{o}_y + 2z\bar{o}_z + \bar{r}^2 - \bar{o}_x^2 - \bar{o}_y^2 - \bar{o}_z^2,$$
$$s.t. \ a\bar{o}_x + b\bar{o}_y + c\bar{o}_z + d = 0, \tag{7}$$

where $\{a, b, c\}$ is the normal vector of plane $\alpha$ and $d$ is the constant term of normal plane. In this equation, the terms $x^2 + y^2 + z^2$, $2x$, $2y$ and $2z$ are regarded as observation values, and meanwhile the variables $\bar{o}_x$, $\bar{o}_y$, $\bar{o}_z$ and $\bar{r}^2 - \bar{o}_x^2 - \bar{o}_y^2 - \bar{o}_z^2$ are regarded as estimated values. We assume that the extracted cross section has $m$ observation points. Equation (7) can be written as the matrix form:

$$Y + e = (B + E_B)A$$
$$s.t. \ KA = K_0 \tag{8}$$

where, $Y = \begin{bmatrix} x_1^2 + y_1^2 + z_1^2 \\ x_2^2 + y_2^2 + z_2^2 \\ \vdots \\ x_m^2 + y_m^2 + z_m^2 \end{bmatrix}$, $B = \begin{bmatrix} 2x_1 & 2y_1 & 2z_1 & 1 \\ 2x_2 & 2y_2 & 2z_2 & 1 \\ \vdots & \vdots & \vdots & \vdots \\ 2x_m & 2y_m & 2z_m & 1 \end{bmatrix}$, $A = \begin{bmatrix} \bar{o}_x \\ \bar{o}_y \\ \bar{o}_z \\ \bar{r}^2 - \bar{o}_x^2 - \bar{o}_y^2 - \bar{o}_z^2 \end{bmatrix}$, $K = [a, b, c, 0]$

and $K_0 = -d$. Parameter $e$ and $E_B$ are the error matrices of $Y$ and $B$. Based on Equation (8), two new matrices $Y_\epsilon = \left[Y^T, K_0\right]^T$ and $B_\epsilon = \left[B^T, K^T\right]^T$ can be generated. Then, we use the SVD technique [23] mentioned in Section 2.2.3 to decompose an augmented matrix $[B_\epsilon, Y_\epsilon]$ to estimate the parameters of the fitted circle defined in matrix $A$.

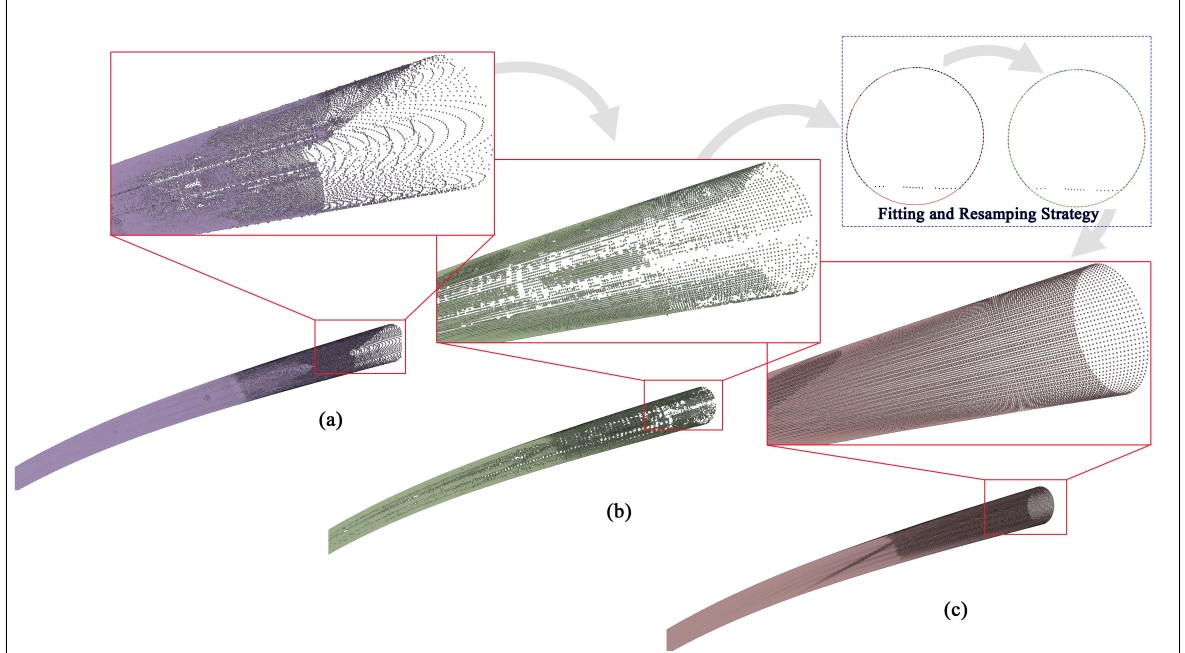

**Figure 8.** Comparisons of tunnel cross-sectional points before and after refinement. (**a**) the raw tunnel point clouds; (**b**,**c**) are results of tunnel cross-sectional points before and after refinement.

### 2.3.4. Continuous Extraction of Tunnel Cross Sections

Once we determine the individual tunnel cross section, the same method can be extended to the whole tunnel to extract the continuous tunnel cross sections. More specifically, we can use an integral formula Equation (9) to repeatedly determine the accurate location $x_{p_d}$ of the next point $p_d = \{x_{p_d}, y_{p_d}, z_{p_d}\}$ along the tunnel axis at a regular interval $\Delta d$ and a starting point $p_s = \{x_{p_s}, y_{p_s}, z_{p_s}\}$:

$$\Delta d = \int_{x_{p_s}}^{x_{p_d}} \sqrt{1 + f'_{xy}(x,y)^2 + f'_{xz}(x,z)^2} d_x. \tag{9}$$

When we get the value of $x_{p_d}$, we substitute it into Equation (1) to get the other two elements of $y_{p_d}$ and $z_{p_d}$ of point $p_d$. This procedure executes repeatedly until all of the locations of the tunnel axis have been calculated. At these locations, the method described in Section 2.3.3 is used to extract their associated tunnel cross-sectional points.

## 3. Performance Evaluation and Discussion

### 3.1. Dataset Specifications

We use Terrestrial Laser Scanner FARO® Focus$^{3D}$ X 330 (Lake Mary, FL, USA) to scan Nanjing metro line S3 from Tiexin bridge to Chunjiang road. The entire scene has five scans with a total length of 127 m and a total number of 200 million points. All the scans are registered into a common reference frame, i.e., coordinate system based on sphere targets. The entire preprocessing procedure relies on FARO processing and registration software SCENE (SCENE 5.5, FARO Technologies Inc., Lake Mary, FL, USA) (https://www.faro.com/products/product-design/faro-scene/). The range of FARO® X 330 is from 0.6 m to 330 m. We acquire five scans of data where we set the distance between any adjacent scans about 23 m to maintain the point density. The point density, i.e., spacing of point clouds is highly dependent on scan resolution. In the case of the FARO® Focus$^{3D}$ X 330 instrument used herein, the scan resolution is expressed as a fraction and the user can choose among 1/1, 1/2, 1/4, 1/5, 1/8, 1/10, 1/16, 1/20 and 1/32, with higher resolution requiring higher scanning time. In our case, we set 1/4, taking 10–15 min for each scan and having 6.136 mm point density at approximately 10 m. In practice, the optimal resolution setting should make a trade-off between the required density points per square meter and the scan duration [25]. To decrease the missing data and data occlusion, we set the field of view zenith angle range from –60° to 90° and azimuth angle range from 0° to 360°. The range error is defined as a systematic measurement error 10 m and 25 m, and one sigma is 0.3 mm under a constraint of a 90% reflective surface. Through an experiment of a fitting flat indoor façade, we find that the precision of the scanner is around 1.5 mm at approximately 10 m scanning ranges.

### 3.2. Geometric Error Evaluation

We evaluate our tunnel cross sections from three aspects including geometry, topology and LoDs. From the perspective of geometry, we analyze the deviations of radii of the extracted cross sections and the chord length based on a series of the central angles, and calculate the nearest Euclidean distance from the cross-sectional points to their nearest raw point clouds.

To evaluate the accuracy of the cross-sectional points, we first analyze the deviations of radii of 1258 cross sections with an interval of 0.1 m and compare them to the designed radius. The mean value and the standard deviation of the radii of these cross sections are 2.7492 m and 1.4 mm, respectively. The statistic results are shown in Figure 9a. Compared with the designed tunnel radius of 2.75 m, the RMSEbetween the extracted radii of tunnel and the designed radius is only 1.6 mm. Considering the influence of the precision of the laser scanner, we select a very flat indoor façade to calibrate the precision of the laser scanner at a distance of 10 m. We fit the flat façade and calculate the mean residual of the fitting error of 1.5 mm between the raw point clouds and the fitted façade. This value is between

1.4 mm and 1.6 mm, which implies that the radii of the extracted cross sections are very stable and accurate compared with the design radius of 2.75 m.

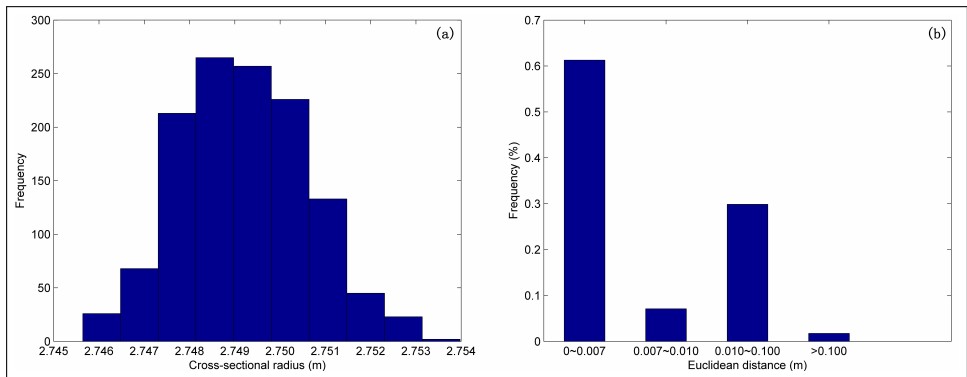

**Figure 9.** Histogram of cross-sectional radii and minimum Euclidean distance from the cross section points to their nearest raw point clouds. (**a**,**b**) represent the statistics of cross-sectional radius and minimum Euclidean distance, respectively.

We also analyze the minimum Euclidean distance from each cross-sectional point to its nearest raw point clouds to inform the "changed" initial raw point clouds. To this end, we divide the horizontal coordinates of the minimum Euclidean distance shown in Figure 9b into four categories: >0.100 m, 0.010–0.100 m, 0.007–0.010 m and 0–0.007 m. The cross section points from the first category only take up to 2%, which is mainly because of the missing data at the vicinity of tunnel track. This missing data arises from occlusion of the tunnel track having higher elevation than its associated rail slab. Because of missing data, the generated cross-sectional points are far away from its nearest raw point clouds, causing the deviation over 0.1 m, as is evident in the Figure 10a. In Figure 10b, the cross-sectional points from the second category account for 30%, which mainly arises from the tunnel bottom and the bolts used to fix the tunnel linings and other installations, e.g., bolts, pipes and wires. The third category accounts for 7% of the cross-sectional points created. From the distribution in Figure 10c, we can see that these points are scattered far away from their corresponding scanners denoted by the blue points in Figure 10. In these areas, the raw point clouds have low density; therefore, the local fitted quadric surfaces are not as accurate as the fitted surfaces derived from the high density of points. Furthermore, some installations and other equipment attached to the tunnel linings can also result in the large deviations. The last category takes up to 61% of the extracted cross-sectional points. The detailed distribution of those point clouds is shown in Figure 10d, where it can be seen that most of the cross-sectional points are distributed approximately 10 m away from their associated scanner. From FARO® Focus$^{3D}$ X 330 scan data processing and registration software SCENE, we obtain the spacing among point clouds at approximately 10 m is 6.136 mm, which makes it consistent with a maximum of 7 mm and explains why these points are distributed well in the vicinity of the scanner around 10 m.

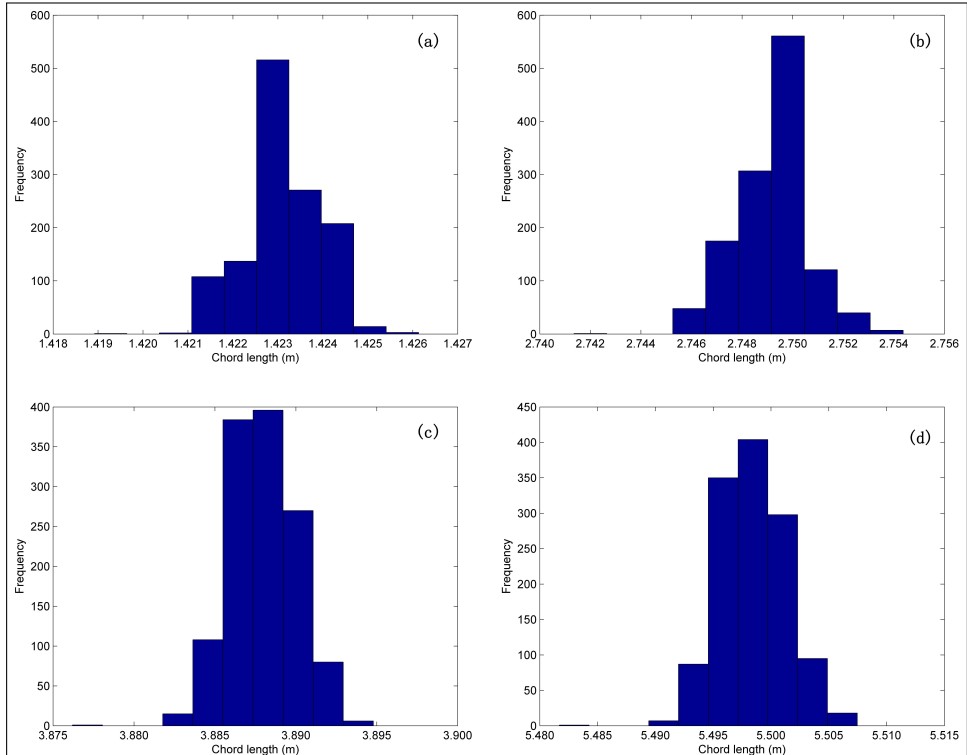

**Figure 10.** Histogram of chord length. (**a**–**d**) are histograms of different chord lengths with respect to central angle of 30°, 60°, 90° and 180°, respectively.

To further evaluate the geometric accuracy of 1258 cross sections, four types of chord lengths associated with central angles of 30°, 60°, 90° and 180° are compared and evaluated. The histogram and statistics of chord length corresponding to the different angles above are shown in Figure 11 and Table 1. It can be clearly seen that the standard deviations $\mathcal{SD}$ and RMSE with respect to columns Mean and *Ref* are monotonically increasing, reaching maximum 2.8 mm and 3.2 mm, respectively. This implies that the errors of cross sections become prominent when the chord length gradually increases. Note that the standard deviation of the chord length with a central angle of 180° reaches the maximum value of 2.8 mm, which is exactly twice the number of standard deviations of the radius (1.4 mm). This means that the deviation of chord length with a center angle of 180° is consistent with twice the deviation of the radius. In addition, compared with the references of chord length calculated using the design radius of 2.75 m, the deviation of the chord length is extremely small, just reaching the maximum of 3.2 mm. Therefore, we can make a safe conclusion that our created cross sections have high geometric accuracy by indirectly evaluating four levels of chord lengths.

**Table 1.** Chord length statistics for 1258 cross sections. The symbols *Min*, *Max*, Mean and $\mathcal{SD}$ represent the minimum, maximum, mean and standard deviation of chord length. '*Ref*' means the reference (standard) of chord length calculated using the design radius of 2.75 m. '$\mathcal{SD}$' and 'RMSE' denote the standard deviations which are calculated using the numbers in the columns 'Mean' and '*Ref*', respectively.

| Central Angles | Min | Max | Mean | Ref | $\mathcal{SD}$ | RMSE |
|---|---|---|---|---|---|---|
| 30° | 1.4189 m | 1.4261 m | 1.4231 m | 1.4235 m | 0.0008 m | 0.0009 m |
| 60° | 2.7414 m | 2.7544 m | 2.7492 m | 2.7500 m | 0.0015 m | 0.0017 m |
| 90° | 3.8762 m | 3.8948 m | 3.8880 m | 3.8891 m | 0.0020 m | 0.0023 m |
| 180° | 5.4817 m | 5.5075 m | 5.4984 m | 5.5000 m | 0.0028 m | 0.0032 m |

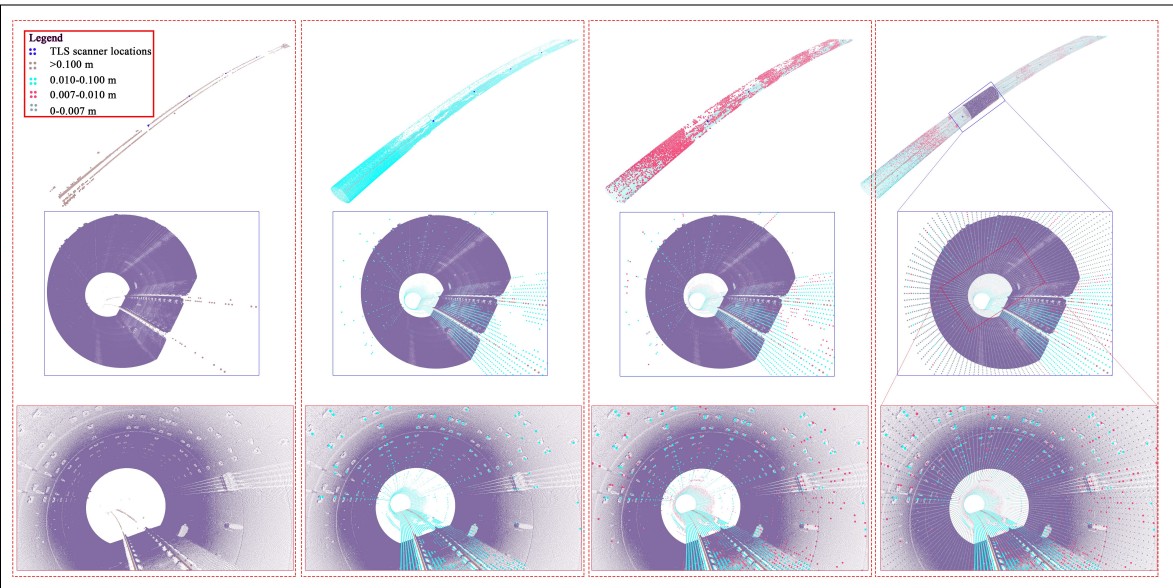

**Figure 11.** The evaluation of cross-sectional points through the measure of minimum Euclidean distance. The cross-sectional points are rendered by the different colors according to the deviations from their raw point clouds. To clearly show where the deviation arises from, the different categories are gradually superimposed onto their raw point clouds, denoted by purple.

### 3.3. Topology Evaluation and LoD Representation

In the field of computational geometry, topology is a crucial index to determine whether the tunnel can be reconstructed via a series of cross sections. Fortunately, our methodology has the capability to maintain the topology of cross sections in two aspects: (1) For each cross section, the tunnel points are generated by sequentially intersecting a straight line with its local fitted quadric surface. Because of this, the generated points within each cross-sectional plane have topological relationships, i.e., clockwise or counter-clockwise order. Using this type of relation, the geometric shape of a specific cross section can be created. (2) For extraction of multiple cross sections (see Section 2.3.4), we also sequentially generate them along the tunnel axis once given an explicit split interval, thereby forming a series of continuous cross sections. That is, the relationship like the former and later cross sections of the currently processing one is considered, and thus the points from any two arbitrary cross sections have one-to-one correspondence. Using the relationship among multiple cross sections, the entire geometric tunnel shape is expected to be generated. As shown in Figure 12a, the purple triangles shown in the enlarged view are the wire meshes of the tunnel that are waved according to the two topologies above.

Because we can simultaneously maintain the point order within each cross section and the order of multiple cross sections, which lays a solid foundation for LoD representation of the metro tunnel. That is, we can use any arbitrary number of cross sections to achieve the abstraction/representation of metro tunnel. Low levels of LoDs with fewer cross sections are lightweight and suitable for storage, web transmission, acceleration of rendering and other tunnel-related applications. In contrast, although the high level of LoDs are not as compact as the shapes with low level LoDs, the geometric accuracy is relatively high. In practice, selection of the optimal LoD is often determined by factors including data acquisition cost, time, accuracy and labor investment and specific tunnel-related applications. For example, when the tunnel models are used in equipment installation on the tunnel linings, the relatively simple tunnel models generated by the sparse cross sections can meet the requirements. When considering tunnel profile deformation, crack and defect detection, and/or water leakage detection, the most detailed tunnel models with high geometric accuracy are expected to meet particular project needs. Furthermore, in the area of tunnel interior visualization, it often needs tunnel models at different LoDs rather than a single LoD. In this case, a multiple LoDs should be generated on the fly to enhance the users' experience. As shown in Figure 12a–g, the tunnel model is rendered

by different LoDs corresponding to profiles of 0.2 m, 0.5 m, 2.0 m, 4.0 m, 6.0 m, 8.0 m and 10.0 m, respectively.

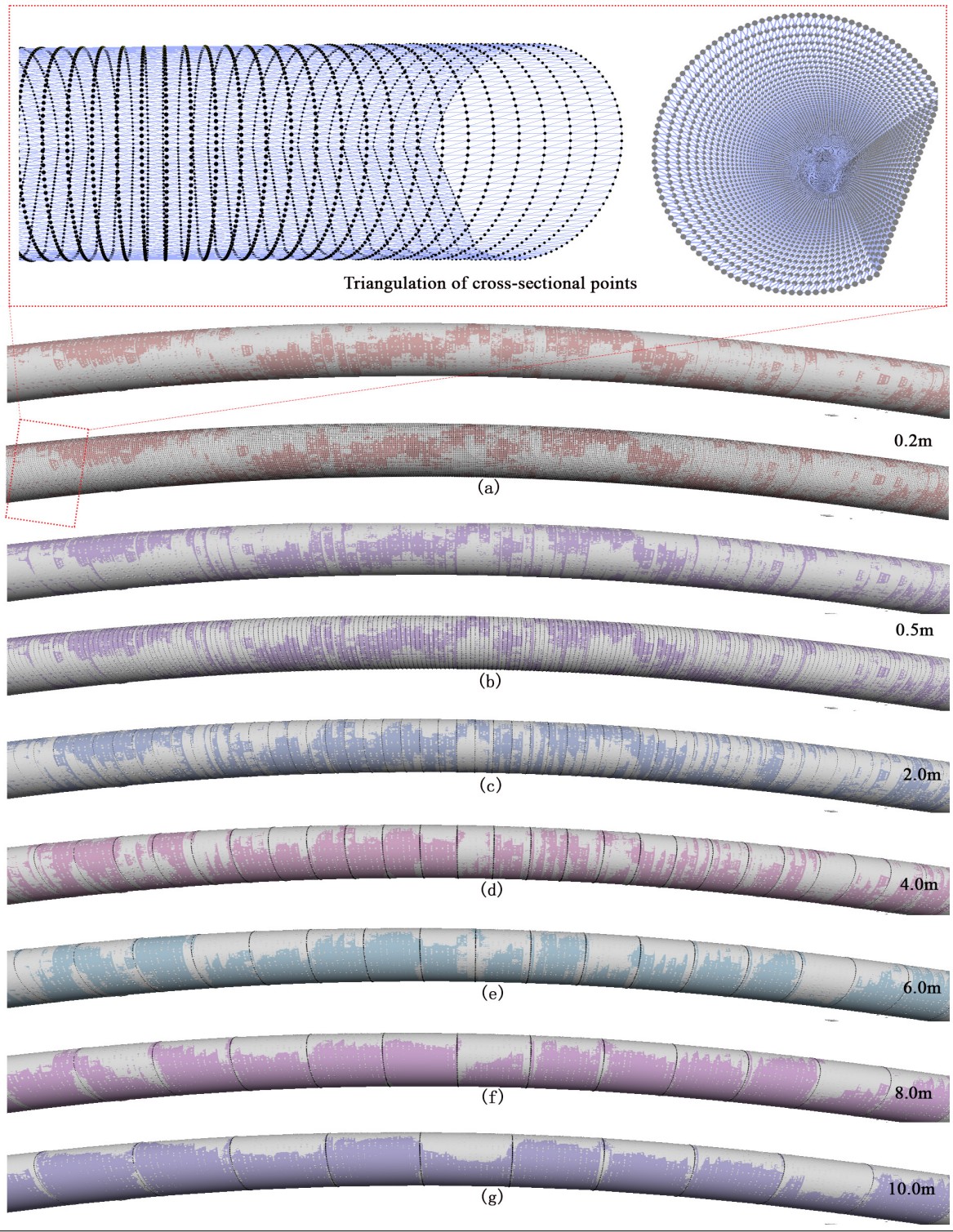

**Figure 12.** The geometric tunnel representation at different LoDs (Level of Details). (**a–g**) are the created tunnel models using the extracted cross sections at the split intervals of 0.2 m, 0.5 m, 2.0 m, 4.0 m, 6.0 m, 8.0 m and 10.0 m. Note that the created tunnel models are superimposed onto their 'reference model' denoted by the gray color. The reference model is produced by a screened Poisson surface reconstruction algorithm through using the raw tunnel point clouds.

To get an optimal split interval $\Delta d$, we first define the measures on two tunnel meshes $\mathcal{X}$ and $\mathcal{Y}$ to evaluate the similarity of two geometric shapes. We first use a Monte Carlo sampling method [26] to obtain a set of points over $\mathcal{X}$ and, for each point on $\mathcal{X}$, it searches the closest point (face or edge) on the other mesh $\mathcal{Y}$. By this way, we can calculate the Mean and RMSE between the sampling points on the sampled mesh $\mathcal{X}$ and their closest points on the target mesh $\mathcal{Y}$. These two measures are regarded as the criterion to determine the similarity of two shapes. In our case, $\mathcal{X}$ represents a tunnel model constructed by a series of profiles at a specific interval. A target mesh $\mathcal{Y}$ denotes a tunnel model that is created by the screened Poisson surface reconstruction algorithm [27] using the raw point clouds. As shown in Figure 12a–g, the target tunnel meshes denoted by gray color are superimposed onto a series of sampled tunnel models with different colors. Through the superimposition, we qualitatively evaluate the similarity by visually inspecting the differences of these two shapes. Meanwhile, we also use the two well accepted statistical measures above to make quantitative evaluations. To this end, we first fix the rotation angle interval $\Delta \varphi$ with a specific value to analyze the variations of Mean and RMSE of the generated tunnel models with respect to the split intervals at 0.2 m, 1.0 m, 2.0 m, 4.0 m, 6.0 m, 8.0 m and 10.0 m. The results are shown in Figure 13, and we unexpectedly discover that, for each polyline corresponding to a specific rotation angle, the Mean and RMSE first decrease and then increase with increasing split interval. This means that the high density of profiles does not guarantee producing a tunnel model with high geometric accuracy. Actually, in our scenario, the split interval of 4.0 m can get the tunnel model with the minimum Mean and RMSE errors. On the contrary, we fix the split interval with a specific value and compare geometric variations with regard to the different rotation angle intervals $\Delta \varphi$, i.e., 3°, 6°, 9° and 12°. The results are shown in Figure 14, where we find that the geometric errors are substantially increasing with the increased rotation angles. The bottom polyline in yellow in Figure 14a corresponds to the tunnel model at a 4.0 m split interval. Therefore, we can make a safe conclusion that extraction of profiles at the space of 4.0 m can guarantee the reconstructed tunnel model with the minimum geometric error. However, in practice, the user should totally make a trade-off between the geometric accuracy, compactness and the specific applications to finally determine how many profiles are needed.

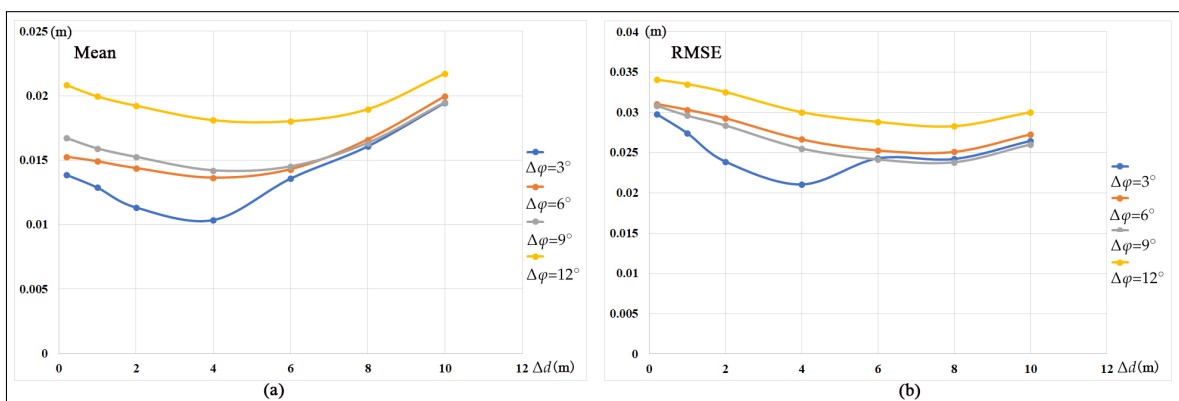

**Figure 13.** The comparisons of those generated and the reference tunnel model at different $\Delta d$ and $\Delta \varphi$.

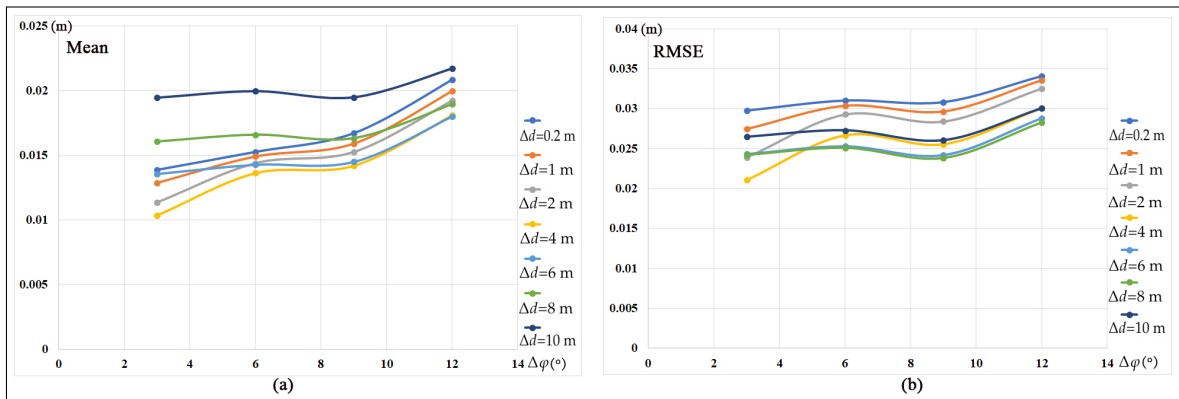

**Figure 14.** The comparisons of those generated and the reference tunnel model at different $\Delta\varphi$ and $\Delta d$.

*3.4. Comparison*

To further evaluate the accuracy of the proposed method, we compare our algorithm with the algorithm proposed in [1] by using the same dataset, i.e., the metro tunnel in Shanghai, China. The Shanghai dataset is described in Table 2. From the Shanghai dataset, we extract 260 tunnel profiles at the fixed split interval of 0.2 m and compare the variation of the fitted radii with the design radius of 2.75 m. Note that the design radii of the tunnels in the Nanjing and Shanghai datasets are both 2.75 m. The detailed statistics for the radii are shown in Table 3. From this table, the radius difference, i.e., RMSE between the cross sections and the designed value is only 4.6 mm, which outperforms the value of 6 mm estimated in Kang's method [1].

**Table 2.** Description of Shanghai subway tunnel dataset. #Pts represents the number of points of the Shanghai metro tunnel scan dataset.

| Scanner | Scan Angular Resolution | #Pts | Range Accuracy |
|---|---|---|---|
| RIEGL VZ-400 | 0.046° | 2,686,866 | ± 5 mm |

**Table 3.** Tunnel radius statistics for 260 cross sections from one scan of Shanghai dataset. #Profile represents the number of the extracted cross sections from Shanghai dataset.

| #Profile | *Max* | *Min* | Mean | $\mathcal{SD}$ | RMSE |
|---|---|---|---|---|---|
| 260 | 2.7632 m | 2.7481 m | 2.7538 m | 0.0026 m | 0.0046 m |

Compared with the Nanjing dataset, the range of the zenith angle for acquiring the Shanghai dataset is relatively narrow, causing very severe missing data at the top and bottom of scanner locations, as evident in Figure 15a,b. Although we suffer from very serious missing data, our algorithm can infer the accurate locations of cross section points in these missing data areas by an assumption of circular tunnel shapes, as proved in Figure 15d,e. Moreover, the Shanghai dataset used in our paper only includes one scan data without overlapping point clouds from other scans. Therefore, the point clouds that are far away from the scanner have low point density. As shown in Figure 15c, the distribution of point clouds from one terminal is very sparse. However, in this case, our algorithm is still insensitive to this adverse situation and has the capability to acquire the credible cross sections (see Figure 15f). This is mainly due to the adaption of "self-adaptation" strategy, which dynamically determines the size of local points to finalize the local quadric surface fitting according to point density variation.

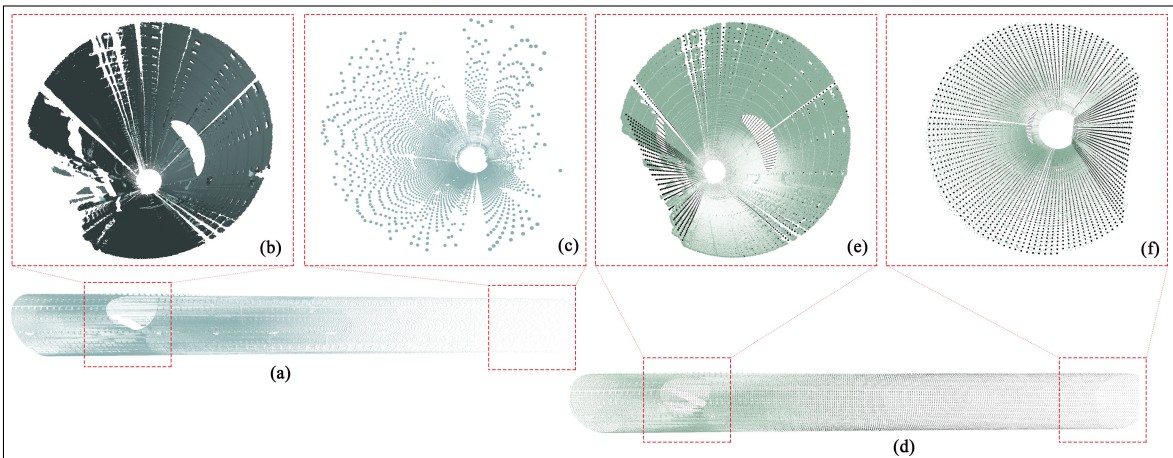

**Figure 15.** Tunnel profile extraction from one scan of Shanghai metro tunnel dataset. The raw tunnel point clouds are shown in (**a**), where large-area of missing data and severe variations of point density are shown in the enlarged view of (**b**,**c**). The extracted 260 metro tunnel profiles, denoted by the black points, are overlaid onto the raw data. From the zoomed views (**d**–**f**), we can vividly see that the missing data areas are filled by the extracted profiles, and meanwhile the profiles are successfully derived from extremely sparse point clouds.

## 4. Conclusions

In this paper, we propose an efficient framework to extract tunnel cross sections from the Terrestrial Laser Scanning point clouds. Our framework starts with the extraction of the tunnel axis using a robust slice-based method. To get an accurate position of the tunnel profiles, we further divide the initial axis into linear and nonlinear segments via an enhanced RANSAC algorithm. In contrast with the traditional RANSAC algorithm, our algorithm has been implemented in the tangent space rather than the data space. By this way, the simultaneous linear and nonlinear axis segmentation problem is thus translated into sole linear element segmentation. It does not only reduce the complex of segmentation but also increases the credibility of segmented results. Based on the segmented axis segments, we can fit the linear and nonlinear segments via the linear regression model and the polynomial regression model, respectively. The extracted attitude of the three-dimensional tunnel axis lays a solid foundation for accurate extraction of tunnel cross sections. The initial cross-sectional points are generated by intersecting a sequence of straight lines that rotate orthogonally around the tunnel axis with the corresponding local quadric surfaces. It should be noted that the size of the quadric surface is not a constant but progressive adjustment according to the density of the local point clouds. This advantage helps the algorithm to adapt to the variations of point clouds and makes the algorithm insensitive to the point density. The initial cross-sectional points are further refined and densified via a "fitting-and-resampling" strategy. That is, the coarse cross-section points are first fitted by an assumption of circular tunnel, i.e., using the circular shape/model to fit the cross-sectional points, followed by resampling of the circular shapes. This process significantly enhances the completeness of cross-sectional points, and more importantly makes the proposed algorithm resistant to missing data.

The results on Nanjing dataset show that the RMSE of the fitting accuracy of the cross sections is 1.687 mm. Compared with the design value of 2.75 m, the RMSE of the radii of extracted cross sections is only 1.60 mm. In addition, by trial-and-error experiments, we find that reconstructing the entire metro tunnel model via cross sections evenly spaced at a distance of around 4 m can get the relatively minimum geometric errors. It should be aware that the extracted tunnel profiles have explicit topology information not only existing within each profile but also among multiple profiles. This is beneficial to geometric processing oriented applications such as metro tunnel reconstruction, model editing and retargeting. We also compare our results with the state-of-the-art method proposed by Kang [1] based upon the same dataset, i.e., Shanghai metro tunnel point clouds. It is to be proved that our methodology can cope with large-area missing data and successfully extract the profiles from very

sparse and unevenly distributed point clouds. Meanwhile, based on the Shanghai dataset, the RMSEs of extracted cross sections' radii with regard to the design radius of 2.75 m are only 4.6 mm, which is superior to Kang's method of 6 mm.

Although our algorithm achieves promising results on Nanjing and Shanghai datasets, there are some drawbacks and interesting ideas which can be further explored to extend the research reported in this paper. The consistency in the connection areas between the linear and nonlinear axis segments may not be guaranteed. That is to say, the extracted cross sections are not consistent on the transition areas. In future work, the global optimization of axis segments might be expected to relieve and even probably eliminate the deviations in the overlapping parts of adjacent different axis models. In addition, our tunnel extraction algorithm is based on an assumption that the tunnel cross sections can be fitted by the circular shapes; therefore, our algorithm is not applicable to more complex and atypical tunnels. For this problem, we plan to segment a complex tunnel into different parts and then adopt the concept of "hybrid representation" to refit these parts by combining different models, i.e., planar model, circular model, elliptical model, and other free-form models. Furthermore, we find that the track laid in the interior of the tunnel and other installations will undoubtedly influence the accuracy of tunnel profiles. To solve this problem, we plan to first remove these irrelevant objects using the computer vision-based classification and recognition algorithm [28] and then extract the tunnel cross sections from the remaining point clouds.

**Author Contributions:** Z.C. analyzed the data and wrote the C++ source code. Y.S. and D.C. helped with the project and study design, paper writing, and analysis. Z.Z., F.J., T.Y., S.X., Z.K. and L.Z. helped with field work and data analysis.

**Funding:** This work was supported by the National Natural Science Foundation of China under Grant No. 31770591 and Grant No. 41701533.

**Acknowledgments:** The authors would like to thank Zhizhong Kang for providing the Shanghai metro tunnel dataset for comparison.

**Conflicts of Interest:** The authors declare no conflict of interest.

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
