# Peer review of "A Flexible Architecture for Extracting Metro Tunnel Cross Sections from Terrestrial Laser Scanning Point Clouds"

_remotesensing, doi:10.3390/rs11030297_

Round 1

Reviewer 1 Report

This paper is well written and have a strong understanding of the previous researches. For the more improvement of the paper, the following comments are suggested by the reviewer. Line 153: how do we determine the order of the polynomial? In this case, the question arises that there are possibilities for overfitting the point clouds. For this, it is required to explain more clearly about the order of the polynomial and how can you guarantee not to overfit the point cloud. Line 214: We use two types of geometric tunnel axis based on synthetic points to prove the applicability of the SG algorithm. —> Need to clearly specify the meaning of synthetic points. Why not using the original points? Do the synthetic points mean the points created for simulation to test the algorithms? Line 230: How do you decide the inlier and outlier? To do this, we need the threshold value to distinguish between inlier and outlier. Need to add more explanation about this. Line 259: This model needs to be solved by using Variable in Error model or Total Least Squares Method. Have you compared your results (SVD) with Variable in Error Model or Total Least Squares Method? Line 322: This is same as the reviewer question in Line 259. This model needs to be solved by using Variable in Error model or Total Least Squares Method. Have you compared your results (SVD) with Variable in Error Model or Total Least Squares Method?

Author Response

Response to Reviewers 1 Comments for Manuscript (remotesensing-431351)

A Flexible Architecture for Extracting Metro Tunnel Cross Sections from Terrestrial Laser Scanning Point Clouds

Zhen Cao, Dong Chen *, Yufeng Shi, Zhenxin Zhang, Fengxiang Jin, Ting Yun, Sheng Xu, Zhizhong Kang and Liqiang Zhang

Dear Reviewer,

We would like to thank you for giving us an opportunity to revise and submit our manuscript entitled “A Flexible Architecture for Extracting Metro Tunnel Cross Sections from Terrestrial Laser Scanning Point Clouds”. We sincerely appreciate your careful reviewing and valuable comments which helped us improving our paper substantially. Following your comments, we believe that we have been successful in significantly improving the presentation and the technical content of our paper. In the revised manuscript, we have addressed, point by point, all the requests and comments from the reviewer to eliminate the linguistic, technical, and structural deficiencies of our manuscript. For convenience, the comments of the reviewer are repeated below in light blue color. The modifications made in the revised manuscript are highlighted using the yellow background. The revised manuscript now completely complies with the guidelines of Remote Sensing. We appreciate your time and look forward to getting a response at your earliest convenience.

Comments and Suggestions for Authors:

This paper is well written and have a strong understanding of the previous researches. For the more improvement of the paper, the following comments are suggested by the reviewer.

Response: Thank you for reviewing our manuscript and providing us your valuable comments and suggestions. We believe that your insightful comments helped us to enhance the clarity and readability of the manuscript. We have carefully considered each one of your comments and modified the manuscript accordingly. Please find below the detailed response to your comments.

Point 1: Line 153: how do we determine the order of the polynomial? In this case, the question arises that there are possibilities for overfitting the point clouds. For this, it is required to explain more clearly about the order of the polynomial and how can you guarantee not to overfit the point cloud.

Response 1: Thank you for your insightful comments. Through experimental analysis, we find that the order of the polynomial is closely related to the length and the curvature of nonlinear tunnel segments. Different values of the above two measures need different order of the polynomial. For our dataset, we use a fifth degree polynomial to guarantee the minimum root mean square fitting error. For other dataset, we need to make trial-and-error experiments to determine an optimal polynomial degree. With regard to the problem of overfitting, the reviewer should not be too worried about it because our axis fitting operation is performed after the tunnel axis segmentation. We only impose the fitting operation on the segmented axis segments. That is, the overfitting problem can be significantly reduced and even eliminated by this divide-and-conquer processing.    

Point 2: Line 214: We use two types of geometric tunnel axis based on synthetic points to prove the applicability of the SG algorithm. —> Need to clearly specify the meaning of synthetic points. Why not using the original points? Do the synthetic points mean the points created for simulation to test the algorithms?

Response 2: Thank you for your constructive comments. The Nanjing tunnel dataset used in our paper has a small curvature, which directly results in the curvature of the extracted tunnel axis is not prominent in the projected xy-plane. Because of this, the real tunnel point clouds cannot fully test the applicability of SG filtering algorithm and its influences for the follow-up tunnel axis segmentation algorithm. To solve this issue, based on synthetic point clouds, we use two types of tunnel shapes with more complicated curvature and noises to prove the applicability and full potential of the proposed SG smoothing and the tunnel axis segmentation algorithm.  Meanwhile, we have given the axis segmented results based on the real tunnel point clouds of Nanjing tunnel dataset (see Figure 6). In addition, we also explain why we use the synthetic point for SG filtering algorithm. Kindly see the revised manuscript on page 7, lines 216-220 for more details.

Point 3: Line 230: How do you decide the inlier and outlier? To do this, we need the threshold value to distinguish between inlier and outlier. Need to add more explanation about this.

Response 3: We are sorry for the ambiguous representation. In our enhanced RANSAC algorithm, the nonlinear segments have been translated into the linear segments via data space transformation. In the transformed tangent space, we recognize the most prominent linear segment at each iteration. We only need two parameters to execute the enhanced RANSAC algorithm: (1) the probability P that at least one of the selected axis points does not contain an outlier, and (2) the Euclidean distance ε from a point to the hypothetical linear model that determines the number of points consistent with the linear model in the tangent space. In our case, we set parameter P to 0.99, which ensures the optimal linear model within a specific number of iterations controlled by P can be obtained. The other parameter ε is determined by the definition of the tangent space. We set ε to 0.01. It is to be noted that the parameter of ε in tangent space is dimensionless although in original data space its unit is consistent with the unit of raw point clouds. Please refer to the revised manuscript on page 8, lines 249-256 for more details.

Point 4: Line 259: This model needs to be solved by using Variable in Error model or Total Least Squares Method. Have you compared your results (SVD) with Variable in Error Model or Total Least Squares Method? Line 322: This is same as the reviewer question in Line 259. This model needs to be solved by using Variable in Error model or Total Least Squares Method. Have you compared your results (SVD) with Variable in Error Model or Total Least Squares Method? 

Response 4:  We have not yet made comparisons with Variable in Error and Total least squares models. One of reasons we chose the total least squares method coupled with SVD is the availability of efficient and numerically more robust in the sense of algorithmic implementation. This viewpoint has been proven in work [1]. Thank you for the reviewer’s valuable suggestion, and in our future work we plan to make comparisons with the suggested methods to further enhance the solution of the equations. 

Reference:

[1] Golub GH; Lan Loan FC. An Analysis of the Total Least Squares Problem. SIAM Journal on Numerical Analysis1980, 17,883-893.

In closing, we would like to again thank you for your valuable suggestions and comments which have helped us to significantly improve the technical content and presentation of our paper. We look forward to receiving your decision regarding the publishability of our revised paper.

Dong Chen

College of Civil Engineering, Nanjing Forestry University, No.159 Longpan Rd., Xuanwu Dist., Nanjing 210037, P.R. China

E-mails: [email protected]

Reviewer 2 Report

The authors face the problem of obtaining cross secions from a point cloud in an original and rigurous way. They combine several computations and data management to avoid outliers and find the best adjutsment.

I only ask that they explain with more detail the threshold and probability used in the RANSAC algorithm.

Author Response

Response to Reviewers 2 Comments for Manuscript (remotesensing-431351)

A Flexible Architecture for Extracting Metro Tunnel Cross Sections from Terrestrial Laser Scanning Point Clouds

Zhen Cao, Dong Chen *, Yufeng Shi, Zhenxin Zhang, Fengxiang Jin, Ting Yun, Sheng Xu, Zhizhong Kang and Liqiang Zhang

Dear Reviewer,

We would like to thank you for giving us an opportunity to revise and submit our manuscript entitled “A Flexible Architecture for Extracting Metro Tunnel Cross Sections from Terrestrial Laser Scanning Point Clouds”. We sincerely appreciate your careful reviewing and valuable comments which helped us improving our paper substantially. Following your comments, we believe that we have been successful in significantly improving the presentation and the technical content of our paper. In the revised manuscript, we have addressed, point by point, all the requests and comments from the reviewer to eliminate the linguistic, technical, and structural deficiencies of our manuscript. For convenience, the comments of the reviewer are repeated below in light blue color. The modifications made in the revised manuscript are highlighted using the yellow background. The revised manuscript now completely complies with the guidelines of Remote Sensing. We appreciate your time and look forward to getting a response at your earliest convenience.

Comments and Suggestions for Authors:

The authors face the problem of obtaining cross sections from a point cloud in an original and rigorous way. They combine several computations and data management to avoid outliers and find the best adjustment.

Response: Thank you for reviewing our manuscript and providing us your insightful comments and suggestions. We have carefully considered each one of your comments and have given the detailed response to your comments. We appreciate your time and look forward to receiving a response at your earliest convenience.

Point 1: I only ask that they explain with more detail the threshold and probability used in the RANSAC algorithm.

Response 1: We are sorry for the ambiguous representation. In the enhanced RANSAC algorithm, the nonlinear tunnel axis segments have been translated into the linear segments via data space transformation. In the transformed tangent space, we recognize the most prominent linear segment at each iteration. We only need two parameters to execute the enhanced RANSAC algorithm: (1) the probability P that at least one of the selected axis points does not contain an outlier, and (2) the Euclidean distance ε from a point to the hypothetical linear model that determines the number of points consistent with the linear model in the tangent space. In our case, we set parameter P to 0.99, which ensures the optimal linear model within a specific number of iterations controlled by P can be obtained. The other parameter ε is determined by the definition of tangent space. We set ε to 0.01. It is to be noted that the parameter of ε in tangent space is dimensionless although in original data space its unit is consistent with the unit of raw point clouds. Kindly see the revised manuscript on page 8, lines 249-256 for more details.

In closing, we would like to again thank you for your valuable suggestions and comments which have helped us to significantly improve the technical content and presentation of our paper. We look forward to receiving your decision regarding the publishability of our revised paper.

Dong Chen

College of Civil Engineering, Nanjing Forestry University, No.159 Longpan Rd., Xuanwu Dist., Nanjing 210037, P.R. China

E-mails: [email protected]

Reviewer 3 Report

Dear authors,

I reviewed you paper titled (A Flexible Architecture for Extracting Metro Tunnel Cross Sections from Terrestrial Laser Scanning Point Clouds) and I found it scientifically significant. I still have the following comments which you can follow to improve:

line 135: you need to define your coordinate axis/system before proceeding in the text. so, as an example" you describe the x axis to be in the direction of travel etc.

line 183: this is depending on the tunnel shape and not generic?

equation 1: So, what does separation into xy and xz brings instead of dealing with the 3D coordinates altogether?

line 162: According to the planning requirements regarding tunnel safety, the tunnel…

is there a reference to this specification?

line 164: However, we use the least squares circular fitting

So, not a higher order polynomial fit?

line 166: I don’t think numbering style is correct!

line 257: can be solved by solving.. please rephrase

line 259: I think it’s better to add a reference to this mathematical solution.

line 300: … classical RANSAC.. better to say conventional.

line 301: to fit cylinder

to fit a cylinder

line 301: …whose parameters include an axis of a cylinder,…

you mean the cosine direction?

line 303: It should be aware that for each straight line, two intersection points can be obtained but one of them is a pseudo point.

it’s useful to show in the figure

figure 7b: annotate the red and black circles in the figure.

line 311: ..or probably because the number of points in Qk is not sufficient to fit the cylinder surface

this is doubtful since we only need 7 points at a minimum.

line 326:

then why not skip the cylinder fitting and use only 3D circle fitting?

line 327: you need a reference for the fitting equations.

line 349:    trade-off between the highest number of points

you mean a tradeoff between the required density points/m2 and the scan duration.

line 366: … and calculate the residual of the fitting error

mean residual or standard deviation of residuals?

line 389: ….we obtain the density of point clouds at approximately 10 m is 6.136 mm

this is not the density, but the spacing between points.

density is measured by points/unit sq. area

line 391:

is there a separation distance between these sections? did you mention the tunnel length at all?

paragraph starting at line 397: English language. the description is unclear to me

Figure 11 caption. (d) are statistics for different chord lengths.

I don’t think this is the proper terminology, may be histogram.

line 430: actual engineering is not a familiar term do you mean actual design?

line 449: surprisedly is not a common word do you mean unexpectedly?

line 525: a typical

Author Response

Response to Reviewers 3 Comments for Manuscript (remotesensing-431351)

A Flexible Architecture for Extracting Metro Tunnel Cross Sections from Terrestrial Laser Scanning Point Clouds

Zhen Cao, Dong Chen *, Yufeng Shi, Zhenxin Zhang, Fengxiang Jin, Ting Yun, Sheng Xu, Zhizhong Kang and Liqiang Zhang

Dear Reviewer,

We would like to thank you for giving us an opportunity to revise and submit our manuscript entitled “A Flexible Architecture for Extracting Metro Tunnel Cross Sections from Terrestrial Laser Scanning Point Clouds”. We sincerely appreciate your careful reviewing and valuable comments which helped us improving our paper substantially. Following your comments, we believe that we have been successful in significantly improving the presentation and the technical content of our paper. In the revised manuscript, we have addressed, point by point, all the requests and comments from the reviewer to eliminate the linguistic, technical, and structural deficiencies of our manuscript. For convenience, the comments of the reviewer are repeated below in light blue color. The modifications made in the revised manuscript are highlighted using the yellow background. The revised manuscript now completely complies with the guidelines of Remote Sensing. We appreciate your time and look forward to getting a response at your earliest convenience.

Comments and Suggestions for Authors:

I reviewed your paper titled (A Flexible Architecture for Extracting Metro Tunnel Cross Sections from Terrestrial Laser Scanning Point Clouds) and I found it scientifically significant. I still have the following comments which you can follow to improve:

Response: Thank you for reviewing our manuscript and providing us your valuable comments and suggestions. We believe that your insightful comments helped us to enhance the clarity and readability of the manuscript. We have carefully considered each one of your comments and modified the manuscript accordingly. Please find below the detailed response to your comments.

Point 1: line 135: you need to define your coordinate axis/system before proceeding in the text. so, as an example" you describe the x axis to be in the direction of travel etc.

Response 1: Thank you for your constructive comments. Actually, we do not require to define the coordinate system before further processing tunnel point clouds. In contrast, we need to determine the extending direction of the tunnel in advance. To this end, we need to calculate the tunnel length along the x, y and z directions. We assume that dx, dy and dz are the tunnel length obtained along x, y and z directions. We can use the criteria given below to determine the extending direction of tunnel:

(1)    If dx>dz and dx>dy, the x-axis can be regarded as the extending direction of the tunnel.

(2)    If dy>dz and dy>dx, the y-axis can be regarded as the extending direction of the tunnel.

(3)    If dx<dz and dx>dy, the y-axis can be regarded as the extending direction of the tunnel.

(4)    If dy<dz and dy>dx, the x-axis can be regarded as the extending direction of the tunnel.

(5)    If dx=dy, either x-axis or y-axis can be considered as the extending direction of the tunnel.   

Once we get the extending direction, in next step we need to slice the tunnel along the extending direction via a proposed slice-based method. For example, if the x-axis is the tunnel extending direction, we just need to project the tunnel points into xy- and xz-planes and get the slices along x-axis from these two projected spaces. Similarly, if the tunnel has the extending direction along the y-axis, we just need to project the tunnel point clouds into xy- and xz-plane and make the slices along y-axis from these projected spaces. No matter which the extending direction is, the procedure of central line extraction is the same. In the revised manuscript, we take the sliced-based method along x-axis as an example to clearly show how to extract the initial tunnel central line from two projected xy- and xz-spaces.

Point 2: line 153: this is depending on the tunnel shape and not generic?

equation 1: So, what does separation into xy and xz brings instead of dealing with the 3D coordinates altogether?

Response 2: Thank you for your insightful comments. The central line segments’ equations are generic for circular tunnels. For example, if the slop of the central line segment in the tangent space is not approximately equal to zero, we use the polynomial regression model to represent the axis segment in the corresponding projected data space. If the slop of the segmented axis is approximately equal to zero in the tangent space, we simply use the linear model to represent the axis segment in the corresponding projected data space. If a slop in one tangent space transformed from the xy-plane is approximately equal to zero but not in another tangent space transformed from the xz-plane, we use the hybrid model by combining polynomial and linear regression models to jointly represent the 3D tunnel segment.

Compared to the conventional central line extraction methods such as cylinder fitting[1] and minimum bounding box[2] methods that piece-wisely extract the 3D central lines from original tunnel point clouds, our projection-based central line extraction method has four advantages: (1) We can get the continuous rather than discrete central line that can be accurately represented by two joint equations. (2) Our method is insensitive to the interior tunnel’s noises and/or outliers. (3) It allows us to use the different applicable equations to fit the central lines, which gives the users’ more flexibility to develop more accurate models. (4) Two separated representations can be regarded as the most simple and straightforward ways to represent 3D tunnel central line. It is well known that the tunnel central line is one of the space curves and it can be represented by intersection of two space surfaces. Thus, the most simple and straightforward way is to project the central line into two orthogonal sub spaces/planes, and get the corresponding linear and/or nonlinear equations to jointly represent the central line equation in 3D space.

References:

[1] X. Xie, X. Lu, Development of a 3D Modeling Algorithm for Tunnel Deformation Monitoring Based on Terrestrial Laser Scanning, Underground Space, 2(1), 16-29, 2017, DOI: http://dx.doi.org/10.1016/j.undsp.2017.02.001.

[2] Shanghai geotechnical engineering investigation and design institute co. LTD. A method of extracting 3D axis of tunnel based on minimum bounding box algorithm, Chinese Patent, CN104392476A, 2015.

Point 3: line 162: According to the planning requirements regarding tunnel safety, the tunnel…

is there a reference to this specification?

Response 3: Yes, the reference has been added in the revised manuscript.

Point 4: line 164: However, we use the least squares circular fitting

So, not a higher order polynomial fit?

Response 4: We are sorry for the inconsistency throughout the manuscript. In our paper, the extracted central line is segmented into linear and nonlinear segments. For the linear segments, we refit them by the linear least squares regression. However, we use a fifth order polynomial model to fit the nonlinear segments. In the revised manuscript, we have corrected this inconsistent representation.

Point 5: line 166: I don’t think numbering style is correct!

Response 5: Thank you for your wonderful observation. In the revised manuscript, we use parentheses to enclose numbers for listed items. We also double check the similar problems and eliminate them throughout the manuscript.

Point 6: line 257: can be solved by solving.. please rephrase

Response 6: We are sorry for the informal usage. We rewrote the corresponding sentences in the revised manuscript. Kindly see the revised manuscript on page 10, line 270 for details.

Point 7: line 259: I think it’s better to add a reference to this mathematical solution.

Response 7: Thank you for your comment. According to the reviewer’s suggestion, we have added two references.

Point 8: line 300: … classical RANSAC.. better to say conventional.

Response 8: We are sorry for the informal usage. In the revised manuscript, it has been substituted by “conventional”.

Point 9: line 301: to fit cylinder

to fit a cylinder

Response 9: We are sorry for the mistake, which has been corrected as per your request.

Point 10: line 301: …whose parameters include an axis of a cylinder,…

you mean the cosine direction?

Response 10:  Yes, the directional cosines of a vector, i.e., the axis of a cylinder are the cosines of the angles between the vector and the three coordinate axes.

Point 11: line 303: It should be aware that for each straight line, two intersection points can be obtained but one of them is a pseudo point.

it’s useful to show in the figure

Response 11: In the revised manuscript, we have enhanced the Figure 7b, where it includes two intersectional points by intersecting a straight line and its local fitted cylinder surface. It should be aware that one of them is a pseudo point, another is a real cross-sectional point.

Point 12: figure 7b: annotate the red and black circles in the figure.

Response 12: Thank you for your wonderful observation. We have given the detailed explanations regarding the red and black circles depicted in Figure 7b. Please refer to the revised manuscript in Figure 7b for more details.

Point 13: line 311: ..or probably because the number of points in Qk is not sufficient to fit the cylinder surface this is doubtful since we only need 7 points at a minimum.

Response 13: Admittedly, for a finite cylinder we just need a total of 7 parameters. A 3D line needs 4 parameters (minimum distance from origin, and 3 parameters for orientation). Then from the point closest to the origin we need 2 distances defining the beginning and end of the cylinder. One more parameter is needed for the radius and we have a 3D cylinder in space defined. But in our context, we mean that the ratio of the number of inlier points in  Qk might be very low. In this case, it is extremely hard to recognize these inliers and fitting cylinder surface fails. To avoid ambiguous statements, we rephrase this sentence. We suggest that the reviewer refer to the revised manuscript on page 11, lines 324-326 for more details. 

Point 14: line 326:

then why not skip the cylinder fitting and use only 3D circle fitting?

Response 14: The local cylinder fitting is used to acquire profile points. As the tunnel point clouds are often contaminated by the noises, outliers and other irrelevant installations, e.g., bolts, pipes and wires, the generated cross-sectional points from a specific profile might incomplete. In this case, we further employ the fitting and resampling strategy via 3D circular model to enhance the coarse tunnel profile and generate the complete profile points.

Point 15: line 327: you need a reference for the fitting equations.

Response 15: Thank you for your comment. In the revised manuscript, we have added a reference as per your suggestion.

Point 16: line 349:    … trade-off between the highest number of points

you mean a tradeoff between the required density points/m2 and the scan duration.

Response 16: Yes, in the practical project, we should make balance between the required density points and the scan duration to improve the efficiency and meanwhile decrease the complexity of the large-scale point cloud management and processing. We are sorry for the misleading usage and in the revised manuscript, the relevant sentence has been rephrased according to your suggestion.

Point 17: line 366: … and calculate the residual of the fitting error

mean residual or standard deviation of residuals?

Response 17: Yes, we use the mean residual of the fitting error of building inner facade to approximately evaluate the precision of the point clouds.

Point 18: line 389: …we obtain the density of point clouds at approximately 10 m is 6.136 mm

this is not the density, but the spacing between points. density is measured by points/unit sq. area

Response 18: We are sorry for the misleading usage. In this context, the term “spacing” seems more appropriate. We have corrected this deficiency in the revised manuscript.

Point 19: line 391: is there a separation distance between these sections? did you mention the tunnel length at all?

 Response 19: We are sorry for our carelessness. In fact, the 1,258 extracted profiles are equally distributed at distance of 0.1 m throughout the tunnel. In the Nanjing dataset used in the manuscript, the total length of the tunnel composited by 5 scans is approximately 127 m. These descriptions have been added in the revised manuscript.    

Point 20: paragraph starting at line 397: English language. the description is unclear to me

Response 20: The corresponding sentence has been rephrased.

Point 21: Figure 11 caption. (d) are statistics for different chord lengths. I don’t think this is the proper terminology, may be histogram.

Response 21: Thank you for your suggestion. The term statistics has been substituted by histogram in the revised manuscript.

Point 22: line 430: actual engineering is not a familiar term do you mean actual design?

Response 22: We rewrote the corresponding sentence in the revised manuscript. Kindly refer to the revised manuscript on page 18, lines 444-446 for more details.

Point 23: line 449: surprisedly is not a common word do you mean unexpectedly?

Response 23: Yes. This deficiency has been corrected as the reviewer’s suggestion.

Point 24: line 525: a typical

Response 24: Here the word “atypical” means unusual and some unique tunnels that cannot be modelled by the circular shapes.

In closing, we would like to again thank you for your valuable suggestions and comments which have helped us to significantly improve the technical content and presentation of our paper. We look forward to receiving your decision regarding the publishability of our revised paper.

Sincerely,

Dong Chen

College of Civil Engineering, Nanjing Forestry University, No.159 Longpan Rd., Xuanwu Dist., Nanjing 210037, P.R. China

E-mails: [email protected]
